# Sparse PCA via Bipartite Matchings

**Megasthenis Asteris**
The University of Texas at Austin
megas@utexas.edu

**Dimitris Papailiopoulos**
University of California, Berkeley
dimitrisp@berkeley.edu

**Anastasios Kyrillidis**
The University of Texas at Austin
anastasios@utexas.edu

**Alexandros G. Dimakis**
The University of Texas at Austin
dimakis@austin.utexas.edu

## Abstract

We consider the following multi-component sparse PCA problem: given a set of data points, we seek to extract a small number of sparse components with *disjoint* supports that jointly capture the maximum possible variance. Such components can be computed one by one, repeatedly solving the single-component problem and deflating the input data matrix, but this greedy procedure is suboptimal. We present a novel algorithm for sparse PCA that jointly optimizes multiple disjoint components. The extracted features capture variance that lies within a multiplicative factor arbitrarily close to 1 from the optimal. Our algorithm is combinatorial and computes the desired components by solving multiple instances of the bipartite maximum weight matching problem. Its complexity grows as a low order polynomial in the ambient dimension of the input data, but exponentially in its rank. However, it can be effectively applied on a low-dimensional sketch of the input data. We evaluate our algorithm on real datasets and empirically demonstrate that in many cases it outperforms existing, deflation-based approaches.

## 1 Introduction

Principal Component Analysis (PCA) reduces data dimensionality by projecting it onto principal subspaces spanned by the leading eigenvectors of the sample covariance matrix. It is one of the most widely used algorithms with applications ranging from computer vision, document clustering to network anomaly detection (see *e.g.* [1, 2, 3, 4, 5]). Sparse PCA is a useful variant that offers higher data interpretability [6, 7, 8] a property that is sometimes desired even at the cost of statistical fidelity [5]. Furthermore, when the obtained features are used in subsequent learning tasks, sparsity potentially leads to better generalization error [9].

Given a real $n \times d$ data matrix $\mathbf{S}$ representing $n$ centered data points in $d$ variables, the first sparse principal component is the sparse vector that maximizes the explained variance:

$$\mathbf{x}_\star \triangleq \underset{\|\mathbf{x}\|_2=1, \|\mathbf{x}\|_0=s}{\arg\max} \mathbf{x}^\top \mathbf{A} \mathbf{x}, \tag{1}$$

where $\mathbf{A} = 1/n \cdot \mathbf{S}^\top \mathbf{S}$ is the $d \times d$ empirical covariance matrix. Unfortunately, the directly enforced sparsity constraint makes the problem NP-hard and hence computationally intractable in general. A significant volume of prior work has focused on various algorithms for approximately solving this optimization problem [3, 5, 7, 8, 10, 11, 12, 13, 14, 15, 16, 17], while some theoretical results have also been established under statistical or spectral assumptions on the input data.

In most cases one is not interested in finding only the first sparse eigenvector, but rather the first $k$, where $k$ is the reduced dimension where the data will be projected. Contrary to the single-component

problem, there has been very limited work on computing multiple sparse components. The scarcity is partially attributed to conventional wisdom stemming from PCA: multiple components can be computed one by one, repeatedly solving the single-component sparse PCA problem (1) and *deflating* [18] the input data to remove information captured by previously extracted components. In fact, multi-component sparse PCA is not a uniquely defined problem in the literature. Deflation-based approaches can lead to different output depending on the type of deflation [18]; extracted components may or may not be orthogonal, while they may have disjoint or overlapping supports. In the statistics literature, where the objective is typically to recover a "true" principal subspace, a branch of work has focused on the "subspace row sparsity" [19], an assumption that leads to sparse components all supported on the same set of variables. While in [20] the authors discuss an alternative perspective on the fundamental objective of the sparse PCA problem.

We focus on the multi-component sparse PCA problem with disjoint supports, *i.e.*, the problem of computing a small number of sparse components with non-overlapping supports that jointly maximize the explained variance:

$$\mathbf{X}_\star \triangleq \underset{\mathbf{X} \in \mathcal{X}_k}{\arg\max} \, \mathrm{TR}\big(\mathbf{X}^\top \mathbf{A} \mathbf{X}\big), \tag{2}$$

$$\mathcal{X}_k \triangleq \big\{ \mathbf{X} \in \mathbb{R}^{d \times k} : \|\mathbf{X}^j\|_2 = 1, \|\mathbf{X}^j\|_0 = s, \mathrm{supp}(\mathbf{X}^i) \cap \mathrm{supp}(\mathbf{X}^j) = \emptyset, \, \forall j \in [k], i < j \big\},$$

with $\mathbf{X}^j$ denoting the $j$th column of $\mathbf{X}$. The number $k$ of the desired components is considered a small constant. Contrary to the greedy sequential approach that repeatedly uses deflation, our algorithm *jointly* computes all the vectors in $\mathbf{X}$ and comes with theoretical approximation guarantees. Note that even if we could solve the single-component sparse PCA problem (1) exactly, the greedy approach could be highly suboptimal. We show this with a simple example in Sec. 7 of the appendix.

**Our Contributions:**

1. We develop an algorithm that provably approximates the solution to the sparse PCA problem (2) within a multiplicative factor arbitrarily close to optimal. Our algorithm is the first that jointly optimizes multiple components with disjoint supports and operates by recasting the sparse PCA problem into multiple instances of the bipartite maximum weight matching problem.

2. The computational complexity of our algorithm grows as a low order polynomial in the ambient dimension $d$, but is exponential in the intrinsic dimension of the input data, *i.e.*, the rank of $\mathbf{A}$. To alleviate the impact of this dependence, our algorithm can be applied on a low-dimensional sketch of the input data to obtain an approximate solution to (2). This extra level of approximation introduces an additional penalty in our theoretical approximation guarantees, which naturally depends on the quality of the sketch and, in turn, the spectral decay of $\mathbf{A}$.

3. We empirically evaluate our algorithm on real datasets, and compare it against state-of-the-art methods for the single-component sparse PCA problem (1) in conjunction with the appropriate deflation step. In many cases, our algorithm significantly outperforms these approaches.

## 2   Our Sparse PCA Algorithm

We present a novel algorithm for the sparse PCA problem with multiple disjoint components. Our algorithm approximately solves the constrained maximization (2) on a $d \times d$ rank-$r$ Positive Semi-Definite (PSD) matrix $\mathbf{A}$ within a multiplicative factor arbitrarily close to 1. It operates by recasting the maximization into multiple instances of the bipartite maximum weight matching problem. Each instance ultimately yields a feasible solution to the original sparse PCA problem; a set of $k$ $s$-sparse components with disjoint supports. Finally, the algorithm exhaustively determines and outputs the set of components that maximizes the explained variance, *i.e.*, the quadratic objective in (2).

The computational complexity of our algorithm grows as a low order polynomial in the ambient dimension $d$ of the input, but exponentially in its rank $r$. Despite the unfavorable dependence on the rank, it is unlikely that a substantial improvement can be achieved in general [21]. However, decoupling the dependence on the ambient and the intrinsic dimension of the input has an interesting ramification; instead of the original input $\mathbf{A}$, our algorithm can be applied on a low-rank surrogate to obtain an approximate solution, alleviating the dependence on $r$. We discuss this in Section 3. In the sequel, we describe the key ideas behind our algorithm, leading up to its guarantees in Theorem 1.

Let $\mathbf{A} = \mathbf{U}\boldsymbol{\Lambda}\mathbf{U}^\top$ denote the truncated eigenvalue decomposition of $\mathbf{A}$; $\boldsymbol{\Lambda}$ is a diagonal $r \times r$ whose $i$th diagonal entry is equal to the $i$th largest eigenvalue of $\mathbf{A}$, while the columns of $\mathbf{U}$ coincide with the corresponding eigenvectors. By the Cauchy-Schwartz inequality, for any $\mathbf{x} \in \mathbb{R}^d$,

$$\mathbf{x}^\top \mathbf{A}\mathbf{x} = \left\| \boldsymbol{\Lambda}^{1/2}\mathbf{U}^\top \mathbf{x} \right\|_2^2 \geq \left\langle \boldsymbol{\Lambda}^{1/2}\mathbf{U}^\top \mathbf{x},\, \mathbf{c} \right\rangle^2, \quad \forall\, \mathbf{c} \in \mathbb{R}^r : \|\mathbf{c}\|_2 = 1. \qquad (3)$$

In fact, equality in (3) can always be achieved for $\mathbf{c}$ colinear to $\boldsymbol{\Lambda}^{1/2}\mathbf{U}\mathbf{x} \in \mathbb{R}^r$ and in turn

$$\mathbf{x}^\top \mathbf{A}\mathbf{x} = \max_{\mathbf{c} \in \mathbb{S}_2^{r-1}} \left\langle \mathbf{x},\, \mathbf{U}\boldsymbol{\Lambda}^{1/2}\mathbf{c} \right\rangle^2,$$

where $\mathbb{S}_2^{r-1}$ denotes the $\ell_2$-unit sphere in $r$ dimensions. More generally, for any $\mathbf{X} \in \mathbb{R}^{d \times k}$,

$$\mathrm{TR}\left(\mathbf{X}^\top \mathbf{A}\mathbf{X}\right) = \sum_{j=1}^{k} {\mathbf{X}^j}^\top \mathbf{A}\mathbf{X}^j = \max_{\mathbf{C}:\mathbf{C}^j \in \mathbb{S}_2^{r-1} \forall j} \sum_{j=1}^{k} \left\langle \mathbf{X}^j,\, \mathbf{U}\boldsymbol{\Lambda}^{1/2}\mathbf{C}^j \right\rangle^2. \qquad (4)$$

Under the variational characterization of the trace objective in (4), the sparse PCA problem (2) can be re-written as a joint maximization over the variables $\mathbf{X}$ and $\mathbf{C}$ as follows:

$$\max_{\mathbf{X} \in \mathcal{X}_k} \mathrm{TR}\left(\mathbf{X}^\top \mathbf{A}\mathbf{X}\right) = \max_{\mathbf{X} \in \mathcal{X}_k} \max_{\mathbf{C}:\mathbf{C}^j \in \mathbb{S}_2^{r-1} \forall j} \sum_{j=1}^{k} \left\langle \mathbf{X}^j,\, \mathbf{U}\boldsymbol{\Lambda}^{1/2}\mathbf{C}^j \right\rangle^2. \qquad (5)$$

The alternative formulation of the sparse PCA problem in (5) may be seemingly more complicated than the original one in (2). However, it takes a step towards decoupling the dependence of the optimization on the ambient and intrinsic dimensions $d$ and $r$, respectively. The motivation behind the introduction of the auxiliary variable $\mathbf{C}$ will become more clear in the sequel.

For a given $\mathbf{C}$, the value of $\mathbf{X} \in \mathcal{X}_k$ that maximizes the objective in (5) for that $\mathbf{C}$ is

$$\widehat{\mathbf{X}} \triangleq \arg\max_{\mathbf{X} \in \mathcal{X}_k} \sum_{j=1}^{k} \left\langle \mathbf{X}^j,\, \mathbf{W}^j \right\rangle^2, \qquad (6)$$

where $\mathbf{W} \triangleq \mathbf{U}\boldsymbol{\Lambda}^{1/2}\mathbf{C}$ is a real $d \times k$ matrix. The constrained, non-convex maximization (6) plays a central role in our developments. We will later describe a combinatorial $O(d \cdot (s \cdot k)^2)$ procedure to efficiently compute $\widehat{\mathbf{X}}$, reducing the maximization to an instance of the bipartite maximum weight matching problem. For now, however, let us assume that such a procedure exists.

Let $\mathbf{X}_\star$, $\mathbf{C}_\star$ be the pair that attains the maximum in (5); in other words, $\mathbf{X}_\star$ is the desired solution to the sparse PCA problem. If the optimal value $\mathbf{C}_\star$ of the auxiliary variable were known, then we would be able to recover $\mathbf{X}_\star$ by solving the maximization (6) for $\mathbf{C} = \mathbf{C}_\star$. Of course, $\mathbf{C}_\star$ is not known, and it is not possible to exhaustively consider all possible values in the domain of $\mathbf{C}$. Instead, we examine only a finite number of possible values of $\mathbf{C}$ over a fine discretization of its domain. In particular, let $\mathcal{N}_{\epsilon/2}(\mathbb{S}_2^{r-1})$ denote a finite $\epsilon/2$-net of the $r$-dimensional $\ell_2$-unit sphere; for any point in $\mathbb{S}_2^{r-1}$, the net contains a point within an $\epsilon/2$ radius from the former. There are several ways to construct such a net. Further, let $[\mathcal{N}_{\epsilon/2}(\mathbb{S}_2^{r-1})]^{\otimes k} \subset \mathbb{R}^{d \times k}$ denote the $k$th Cartesian power of the aforementioned $\epsilon/2$-net. By construction, this collection of points contains a matrix $\mathbf{C}$ that is column-wise close to $\mathbf{C}_\star$. In turn, it can be shown using the properties of the net, that the candidate solution $\mathbf{X} \in \mathcal{X}_k$ obtained through (6) at that point $\mathbf{C}$ will be approximately as good as the optimal $\mathbf{X}_\star$ in terms of the quadratic objective in (2).

All above observations yield a procedure for approximately solving the sparse PCA problem (2). The steps are outlined in Algorithm 1. Given the desired number of components $k$ and an accuracy parameter $\epsilon \in (0, 1)$, the algorithm generates a net $[\mathcal{N}_{\epsilon/2}(\mathbb{S}_2^{r-1})]^{\otimes k}$ and iterates over its points. At each point $\mathbf{C}$, it computes a feasible solution for the sparse PCA problem – a set of $k$ $s$-sparse components – by solving maximization (6) via a procedure (Alg. 2) that will be described in the sequel. The algorithm collects the candidate solutions identified at the points of the net. The best among them achieves an objective in (2) that provably lies close to optimal. More formally,

**Theorem 1.** *For any real $d \times d$ rank-$r$ PSD matrix $\mathbf{A}$, desired number of components $k$, number $s$ of nonzero entries per component, and accuracy parameter $\epsilon \in (0, 1)$, Algorithm 1 outputs $\overline{\mathbf{X}} \in \mathcal{X}_k$ such that*

$$\mathrm{TR}\left(\overline{\mathbf{X}}^\top \mathbf{A}\overline{\mathbf{X}}\right) \geq (1 - \epsilon) \cdot \mathrm{TR}\left(\mathbf{X}_\star^\top \mathbf{A}\mathbf{X}_\star\right),$$

*where $\mathbf{X}_\star \triangleq \arg\max_{\mathbf{X} \in \mathcal{X}_k} \mathrm{TR}\left(\mathbf{X}^\top \mathbf{A}\mathbf{X}\right)$, in time $T_{SVD}(r) + O\left(\left(\frac{4}{\epsilon}\right)^{r \cdot k} \cdot d \cdot (s \cdot k)^2\right)$.*

Algorithm 1 is the first nontrivial algorithm that provably approximates the solution of the sparse PCA problem (2). According to Theorem 1, it achieves an objective value that lies within a multiplicative factor from the optimal, arbitrarily close to 1. Its complexity grows as a low-order polynomial in the dimension $d$ of the input, but exponentially in the intrinsic dimension $r$. Note, however, that it can be substantially better compared to the $O(d^{s \cdot k})$ brute force approach that

---

**Algorithm 1** Sparse PCA (Multiple disjoint components)

**input** : PSD $d \times d$ rank-$r$ matrix $\mathbf{A}$, $\epsilon \in (0, 1)$, $k \in \mathbb{Z}_+$.
**output** : $\overline{\mathbf{X}} \in \mathcal{X}_k$                     {Theorem 1}
1:   $\mathcal{C} \leftarrow \{\}$
2:   $[\mathbf{U}, \boldsymbol{\Lambda}] \leftarrow \texttt{EIG}(\mathbf{A})$
3:   **for each** $\mathbf{C} \in [\mathcal{N}_{\epsilon/2}(\mathbb{S}_2^{r-1})]^{\otimes k}$ **do**
4:      $\mathbf{W} \leftarrow \mathbf{U}\boldsymbol{\Lambda}^{1/2}\mathbf{C}$          $\{\mathbf{W} \in \mathbb{R}^{d \times k}\}$
5:      $\widehat{\mathbf{X}} \leftarrow \arg\max_{\mathbf{X} \in \mathcal{X}_k} \sum_{j=1}^{k} \langle \mathbf{X}^j, \mathbf{W}^j \rangle^2$    {Alg. 2}
6:      $\mathcal{C} \leftarrow \mathcal{C} \cup \{\widehat{\mathbf{X}}\}$
7:   **end for**
8:   $\overline{\mathbf{X}} \leftarrow \arg\max_{\mathbf{X} \in \mathcal{C}} \text{TR}(\mathbf{X}^\top \mathbf{A} \mathbf{X})$

---

exhaustively considers all candidate supports for the $k$ sparse components. The complexity of our algorithm follows from the cardinality of the net and the complexity of Algorithm 2, the subroutine that solves the constrained maximization (6). The latter is a key ingredient of our algorithm, and is discussed in detail in the next subsection. A formal proof of Theorem 1 is provided in Section 9.2.

## 2.1 Sparse Components via Bipartite Matchings

In the core of Alg. 1 lies a procedure that solves the constrained maximization (6) (Alg. 2). The latter breaks down the maximization into two stages. First, it identifies the support of the optimal solution $\widehat{\mathbf{X}}$ by solving an instance of the maximum weight matching problem on a bipartite graph $G$. Then, it recovers the exact values of its nonzero entries based on the Cauchy-Schwarz inequality. In the sequel, we provide a brief description of Alg. 2, leading up to its guarantees in Lemma 2.1.

Let $\mathcal{I}_j \triangleq \text{supp}(\widehat{\mathbf{X}}^j)$ be the support of the $j$th column of $\widehat{\mathbf{X}}$, $j = 1, \dots, k$. The objective in (6) becomes

$$\sum_{j=1}^{k} \langle \widehat{\mathbf{X}}^j, \mathbf{W}^j \rangle^2 = \sum_{j=1}^{k} \Big( \sum_{i \in \mathcal{I}_j} \widehat{X}_{ij} \cdot W_{ij} \Big)^2 \le \sum_{j=1}^{k} \sum_{i \in \mathcal{I}_j} W_{ij}^2. \tag{7}$$

The inequality is due to Cauchy-Schwarz and the constraint $\|\mathbf{X}^j\|_2 = 1 \,\forall\, j \in \{1, \dots, k\}$. In fact, if an oracle reveals the supports $\mathcal{I}_j$, $j = 1, \dots, k$, the upper bound in (7) can always be achieved by setting the nonzero entries of $\widehat{\mathbf{X}}$ as in Algorithm 2 (Line 6). Therefore, the key in solving (6) is determining the collection of supports to maximize the right-hand side of (7).

By constraint, the sets $\mathcal{I}_j$ must be pairwise disjoint, each with cardinality $s$. Consider a weighted bipartite graph $G = (U = \{U_1, \dots, U_k\}, V, E)$ constructed as follows[1] (Fig. 1):

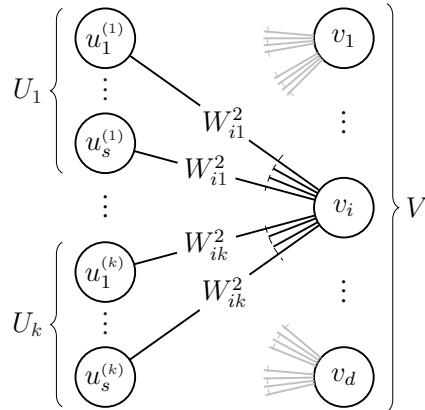

- $V$ is a set of $d$ vertices $v_1, \dots, v_d$, corresponding to the $d$ variables, *i.e.*, the $d$ rows of $\widehat{\mathbf{X}}$.

- $U$ is a set of $k \cdot s$ vertices, conceptually partitioned into $k$ disjoint subsets $U_1, \dots, U_k$, each of cardinality $s$. The $j$th subset, $U_j$, is associated with the support $\mathcal{I}_j$; the $s$ vertices $u_\alpha^{(j)}$, $\alpha = 1, \dots, s$ in $U_j$ serve as placeholders for the variables/indices in $\mathcal{I}_j$.

- Finally, the edge set is $E = U \times V$. The edge weights are determined by the $d \times k$ matrix $\mathbf{W}$ in (6). In particular, the weight of edge $(u_\alpha^{(j)}, v_i)$ is equal to $W_{ij}^2$. Note that all vertices in $U_j$ are effectively identical; they all share a common neighborhood and edge weights.

Figure 1: The graph $G$ generated by Alg. 2. It is used to determine the support of the solution $\widehat{\mathbf{X}}$ in (6).

Any feasible support $\{\mathcal{I}_j\}_{j=1}^k$ corresponds to a *perfect matching* in $G$ and vice-versa. Recall that a matching is a subset of the edges containing no two edges incident to the same vertex, while a perfect matching, in the case of an unbalanced bipartite graph $G = (U, V, E)$ with $|U| \leq |V|$, is a matching that contains at least one incident edge for each vertex in $U$. Given a perfect matching $\mathcal{M} \subseteq E$, the disjoint neighborhoods of $U_j$s under $\mathcal{M}$ yield a support $\{\mathcal{I}_j\}_{j=1}^k$. Con-

---

**Algorithm 2** Compute Candidate Solution

---
**input** Real $d \times k$ matrix $\mathbf{W}$
**output** $\widehat{\mathbf{X}} = \arg\max_{\mathbf{X} \in \mathcal{X}_k} \sum_{j=1}^k \langle \mathbf{X}^j, \mathbf{W}^j \rangle^2$
 1: $G(\{U_j\}_{j=1}^k, V, E) \leftarrow \text{GENBIGRAPH}(\mathbf{W})$   {Alg. 4}
 2: $\mathcal{M} \leftarrow \text{MAXWEIGHTMATCH}(G)$       $\{\subset E\}$
 3: $\widehat{\mathbf{X}} \leftarrow \mathbf{0}_{d \times k}$
 4: **for** $j = 1, \ldots, k$ **do**
 5:    $\mathcal{I}_j \leftarrow \{i \in \{1, \ldots, d\} : (u, v_i) \in \mathcal{M}, u \in U_j\}$
 6:    $[\widehat{\mathbf{X}}^j]_{\mathcal{I}_j} \leftarrow [\mathbf{W}^j]_{\mathcal{I}_j}/\|[\mathbf{W}^j]_{\mathcal{I}_j}\|_2$
 7: **end for**

---

versely, any valid support yields a unique perfect matching in $G$ (taking into account that all vertices in $U_j$ are isomorphic). Moreover, due to the choice of weights in $G$, the right-hand side of (7) for a given support $\{\mathcal{I}_j\}_{j=1}^k$ is equal to the weight of the matching $\mathcal{M}$ in $G$ induced by the former, *i.e.*, $\sum_{j=1}^k \sum_{i \in \mathcal{I}_j} W_{ij}^2 = \sum_{(u,v) \in \mathcal{M}} w(u, v)$. It follows that determining the support of the solution in (6), reduces to solving the maximum weight matching problem on the bipartite graph $G$.

Algorithm 2 readily follows. Given $\mathbf{W} \in \mathbb{R}^{d \times k}$, the algorithm generates a weighted bipartite graph $G$ as described, and computes its maximum weight matching. Based on the latter, it first recovers the desired support of $\widehat{\mathbf{X}}$ (Line 5), and subsequently the exact values of its nonzero entries (Line 6). The running time is dominated by the computation of the matching, which can be done in $O(|E||U| + |U|^2 \log |U|)$ using a variant of the Hungarian algorithm [22]. Hence,

**Lemma 2.1.** *For any $\mathbf{W} \in \mathbb{R}^{d \times k}$, Algorithm 2 computes the solution to* (6)*, in time $O(d \cdot (s \cdot k)^2)$.*

A more formal analysis and proof of Lemma 2.1 is available in Sec. 9.1. This completes the description of our sparse PCA algorithm (Alg. 1) and the proof sketch of Theorem 1.

## 3 Sparse PCA on Low-Dimensional Sketches

Algorithm 1 approximately solves the sparse PCA problem (2) on a $d \times d$ rank-$r$ PSD matrix $\mathbf{A}$ in time that grows as a low-order polynomial in the ambient dimension $d$, but depends exponentially on $r$. This dependence can be prohibitive in practice. To mitigate its effect, we can apply our sparse PCA algorithm on a low-rank sketch of $\mathbf{A}$. Intuitively, the quality of the extracted

---

**Algorithm 3** Sparse PCA on Low Dim. Sketch

---
**input** : Real $n \times d$ $\mathbf{S}$, $r \in \mathbb{Z}_+$, $\epsilon \in (0, 1)$, $k \in \mathbb{Z}_+$.
**output** $\overline{\mathbf{X}}_{(r)} \in \mathcal{X}_k$.            {Thm. 2}
 1: $\overline{\mathbf{S}} \leftarrow \text{SKETCH}(\mathbf{S}, r)$
 2: $\overline{\mathbf{A}} \leftarrow \overline{\mathbf{S}}^\top \overline{\mathbf{S}}$
 3: $\overline{\mathbf{X}}_{(r)} \leftarrow \text{ALGORITHM 1}(\overline{\mathbf{A}}, \epsilon, k)$.

---

components should depend on how well that low-rank surrogate approximates the original input.

More formally, let $\mathbf{S}$ be the real $n \times d$ data matrix representing $n$ (potentially centered) datapoints in $d$ variables, and $\mathbf{A}$ the corresponding $d \times d$ covariance matrix. Further, let $\overline{\mathbf{S}}$ be a low-dimensional sketch of the original data; an $n \times d$ matrix whose rows lie in an $r$-dimensional subspace, with $r$ being an accuracy parameter. Such a sketch can be obtained in several ways, including for example exact or approximate SVD, or online sketching methods [23]. Finally, let $\overline{\mathbf{A}} = 1/n \cdot \overline{\mathbf{S}}^\top \overline{\mathbf{S}}$ be the covariance matrix of the sketched data. Then, instead of $\mathbf{A}$, we can approximately solve the sparse PCA problem by applying Algorithm 1 on the low-rank surrogate $\overline{\mathbf{A}}$. The above are formally outlined in Algorithm 3. We note that the covariance matrix $\overline{\mathbf{A}}$ does not need to be explicitly computed; Algorithm 1 can operate directly on the (sketched) input data matrix.

**Theorem 2.** *For any $n \times d$ input data matrix $\mathbf{S}$, with corresponding empirical covariance matrix $\mathbf{A} = 1/n \cdot \mathbf{S}^\top \mathbf{S}$, any desired number of components $k$, and accuracy parameters $\epsilon \in (0, 1)$ and $r$, Algorithm 3 outputs $\mathbf{X}_{(r)} \in \mathcal{X}_k$ such that*

$$\text{TR}(\mathbf{X}_{(r)}^\top \mathbf{A} \mathbf{X}_{(r)}) \geq (1 - \epsilon) \cdot \text{TR}(\mathbf{X}_\star^\top \mathbf{A} \mathbf{X}_\star) - 2 \cdot k \cdot \|\mathbf{A} - \overline{\mathbf{A}}\|_2,$$

*where $\mathbf{X}_\star \triangleq \arg\max_{\mathbf{X} \in \mathcal{X}_k} \text{TR}(\mathbf{X}^\top \mathbf{A} \mathbf{X})$, in time $T_{SKETCH}(r) + T_{SVD}(r) + O\left(\left(\frac{4}{\epsilon}\right)^{r \cdot k} \cdot d \cdot (s \cdot k)^2\right)$.*

The error term $\|\mathbf{A} - \overline{\mathbf{A}}\|_2$ and in turn the tightness of the approximation guarantees hinges on the quality of the sketch. Roughly, higher values of the parameter $r$ should allow for a sketch that more accurately represents the original data, leading to tighter guarantees. That is the case, for example, when the sketch is obtained through exact SVD. In that sense, Theorem 2 establishes a natural trade-off between the running time of Algorithm 3 and the quality of the approximation guarantees. (See [24] for additional results.) A formal proof of Theorem 2 is provided in Appendix Section 9.3.

## 4  Related Work

A significant volume of work has focused on the single-component sparse PCA problem (1); we scratch the surface and refer the reader to citations therein. Representative examples range from early heuristics in [7], to the LASSO based techniques in [8], the elastic net $\ell_1$-regression in [5], $\ell_1$ and $\ell_0$ regularized optimization methods such as GPower in [10], a greedy branch-and-bound technique in [11], or semidefinite programming approaches [3, 12, 13]. Many focus on a statistical analysis that pertains to specific data models and the recovery of a "true" sparse component. In practice, the most competitive results in terms of the maximization in (1) seem to be achieved by *(i)* the simple and efficient truncated power (TPower) iteration of [14], *(ii)* the approach of [15] stemming from an expectation-maximization (EM) formulation, and *(iii)* the (SpanSPCA) framework of [16] which solves the sparse PCA problem through low rank approximations based on [17].

We are not aware of any algorithm that explicitly addresses the multi-component sparse PCA problem (2). Multiple components can be extracted by repeatedly solving (1) with one of the aforementioned methods. To ensure disjoint supports, variables "selected" by a component are removed from the dataset. However, this greedy approach can result in highly suboptimal objective value (see Sec. 7). More generally, there has been relatively limited work in the estimation of principal subspaces or multiple components under sparsity constraints. Non-deflation-based algorithms include extensions of the diagonal [25] and iterative thresholding [26] approaches, while [27] and [28] propose methods that rely on the "row sparsity for subspaces" assumption of [19]. These methods yield components supported on a common set of variables, and hence solve a problem different from (2). In [20], the authors discuss the multi-component sparse PCA problem, propose an alternative objective function and for that problem obtain interesting theoretical guarantees. In [29] they consider a structured variant of sparse PCA where higher-order structure is encoded by an atomic norm regularization. Finally, [30] develops a framework for sparse matrix factorizaiton problems, based on an atomic norm. Their framework captures sparse PCA –although not explicitly the constraint of disjoint supports– but the resulting optimization problem, albeit convex, is NP-hard.

## 5  Experiments

We evaluate our algorithm on a series of real datasets, and compare it to deflation-based approaches for sparse PCA using TPower [14], EM [15], and SpanSPCA [16]. The latter are representative of the state of the art for the single-component sparse PCA problem (1). Multiple components are computed one by one. To ensure disjoint supports, the deflation step effectively amounts to removing from the dataset all variables used by previously extracted components. For algorithms that are randomly initialized, we depict best results over multiple random restarts. Additional experimental results are listed in Section 11 of the appendix.

Our experiments are conducted in a Matlab environment. Due to its nature, our algorithm is easily parallelizable; its prototypical implementation utilizes the Parallel Pool Matlab feature to exploit multicore (or distributed cluster) capabilities. Recall that our algorithm operates on a low-rank approximation of the input data. Unless otherwise specified, it is configured for a rank-4 approximation obtained via truncated SVD. Finally, we note that our algorithm is slower than the deflation-based methods. We set a barrier on the execution time of our algorithm at the cost of the theoretical approximation guarantees; the algorithm returns the best result at the time of termination. This "early termination" can only hurt the performance of our algorithm.

**Leukemia Dataset.**  We evaluate our algorithm on the Leukemia dataset [31]. The dataset comprises 72 samples, each consisting of expression values for 12582 probe sets. We extract $k = 5$ sparse components, each active on $s = 50$ features. In Fig. 2(a), we plot the cumulative explained variance versus the number of components. Deflation-based approaches are greedy: the leading

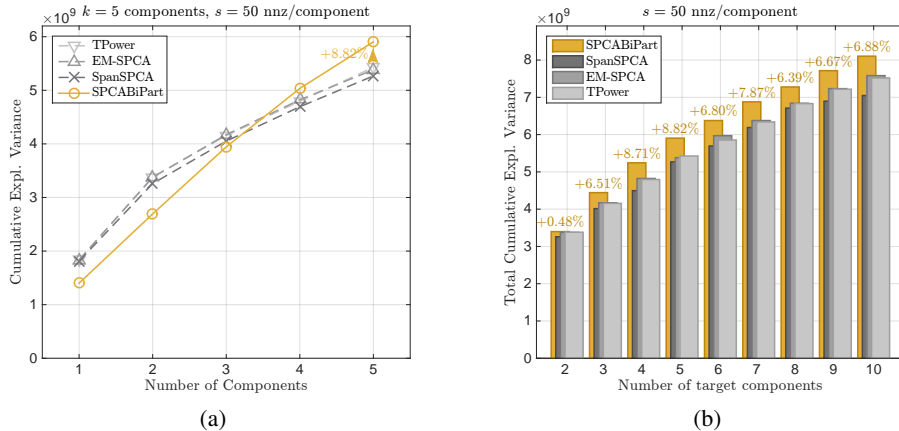

(a)                                                    (b)

Figure 2: Cumul. variance captured by $k$ $s$-sparse extracted components; Leukemia dataset [31]. We arbitrarily set $s = 50$ nonzero entries per component. Fig. 2(a) depicts the cumul. variance vs the number of components, for $k = 5$. Deflation-based approaches are greedy; first components capture high variance, but subsequent contribute less. Our algorithm jointly optimizes the $k$ components and achieves higher objective. Fig. 2(b) depicts the cumul. variance achieved for various values of $k$.

components capture high values of variance, but subsequent ones contribute less. On the contrary, our algorithm jointly optimizes the $k = 5$ components and achieves higher *total* cumulative variance; one cannot identify a top component. We repeat the experiment for multiple values of $k$. Fig. 2(b) depicts the total cumulative variance capture by each method, for each value of $k$.

**Additional Datasets.** We repeat the experiment on multiple datasets, arbitrarily selected from [31]. Table 1 lists the total cumulative variance captured by $k = 5$ components, each with $s = 40$ nonzero entries, extracted using the four methods. Our algorithm achieves the highest values in most cases.

**Bag of Words (BoW) Dataset.** [31] This is a collection of text corpora stored under the "bag-of-words" model. For each text corpus, a vocabulary of $d$ words is extracted upon tokenization, and the removal of stopwords and words appearing fewer than ten times in total. Each document is then represented as a vector in that $d$-dimensional space, with the $i$th entry corresponding to the number of appearances of the $i$th vocabulary entry in the document.

We solve the sparse PCA problem (2) on the word-by-word cooccurrence matrix, and extract $k = 8$ sparse components, each with cardinality $s = 10$. We note that the latter is not explicitly constructed; our algorithm can operate directly on the input word-by-document matrix. Table 2 lists the variance captured by each method; our algorithm consistently outperforms the other approaches.

Finally, note that here each sparse component effectively *selects* a small set of words. In turn, the $k$ extracted components can be interpreted as a set of well-separated *topics*. In Table 3, we list the

|  |  | TPower | EM sPCA | SpanSPCA | SPCABiPart |
|---|---|---|---|---|---|
| AMZN COM REV | (1500×10000) | $7.31e+03$ | $7.32e+03$ | $7.31e+03$ | $\mathbf{7.79e+03}$ |
| ARCENCE TRAIN | (100×10000) | $1.08e+07$ | $1.02e+07$ | $1.08e+07$ | $\mathbf{1.10e+07}$ |
| CBCL FACE TRAIN | (2429×361) | $5.06e+00$ | $5.18e+00$ | $5.23e+00$ | $\mathbf{5.29e+00}$ |
| ISOLET-5 | (1559×617) | $3.31e+01$ | $3.43e+01$ | $3.34e+01$ | $\mathbf{3.51e+01}$ |
| LEUKEMIA | (72×12582) | $5.00e+09$ | $5.03e+09$ | $4.84e+09$ | $\mathbf{5.37e+09}$ |
| PEMS TRAIN | (267×138672) | $\mathbf{3.94e+00}$ | $3.58e+00$ | $3.89e+00$ | $3.75e+00$ |
| MFEAT PIX | (2000×240) | $5.00e+02$ | $5.27e+02$ | $5.08e+02$ | $\mathbf{5.47e+02}$ |

Table 1: Total cumulative variance captured by $k = 5$ 40-sparse extracted components on various datasets [31]. For each dataset, we list the size (#samples×#variables) and the value of variance captured by each method. Our algorithm operates on a rank-4 sketch in all cases.

|  |  | TPower | EM sPCA | SpanSPCA | SPCABiPart |
|---|---|---|---|---|---|
| BOW:NIPS | (1500×12419) | $2.51e+03$ | $2.57e+03$ | $2.53e+03$ | **3.34e+03** (+29.98%) |
| BOW:KOS | (3430×6906) | $4.14e+01$ | $4.24e+01$ | $4.21e+01$ | **6.14e+01** (+44.57%) |
| BOW:ENRON | (39861×28102) | $2.11e+02$ | $2.00e+02$ | $2.09e+02$ | **2.38e+02** (+12.90%) |
| BOW:NYTIMES | (300000×102660) | $4.81e+01$ | − | $4.81e+01$ | **5.31e+01** (+10.38%) |

Table 2: Total variance captured by $k=8$ extracted components, each with $s=15$ nonzero entries – Bag of Words dataset [31]. For each corpus, we list the size (#documents×#vocabulary-size) and the explained variance. Our algorithm operates on a rank-5 sketch in all cases.

topics extracted from the NY Times corpus (part of the Bag of Words dataset). The corpus consists of $3 \cdot 10^5$ news articles and a vocabulary of $d = 102660$ words.

# 6 Conclusions

We considered the sparse PCA problem for multiple components with disjoint supports. Existing methods for the single component problem can be used along with an appropriate deflation step to compute multiple components one by one, leading to potentially suboptimal results. We presented a novel algorithm for jointly computing multiple sparse and disjoint components with provable approximation guarantees. Our algorithm is combinatorial and exploits interesting connections between the sparse PCA and the bipartite maximum weight matching problems. Its running time grows as a low-order polynomial in the ambient dimension of the input data, but depends exponentially on its rank. To alleviate this dependency, we can apply the algorithm on a low-dimensional sketch of the input, at the cost of an additional error in our theoretical approximation guarantees. Empirical evaluation showed that in many cases our algorithm outperforms deflation-based approaches.

**Acknowledgments**

DP is generously supported by NSF awards CCF-1217058 and CCF-1116404 and MURI AFOSR grant 556016. This research has been supported by NSF Grants CCF 1344179, 1344364, 1407278, 1422549 and ARO YIP W911NF-14-1-0258.

## Footnotes

[1]The construction is formally outlined in Algorithm 4 in Section 8.

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

|  | Topic 1 | Topic 2 | Topic 3 | Topic 4 | Topic 5 | Topic 6 | Topic 7 | Topic 8 |
|---|---|---|---|---|---|---|---|---|
| 1: | percent | zzz_united_states | zzz_bush | company | team | cup | school | zzz_al_gore |
| 2: | million | zzz_u_s | official | companies | game | minutes | student | zzz_george_bush |
| 3: | money | zzz_american | government | market | season | add | children | campaign |
| 4: | high | attack | president | stock | player | tablespoon | women | election |
| 5: | program | military | group | business | play | oil | show | plan |
| 6: | number | palestinian | leader | billion | point | teaspoon | book | tax |
| 7: | need | war | country | analyst | run | water | family | public |
| 8: | part | administration | political | firm | right | pepper | look | zzz_washington |
| 9: | problem | zzz_white_house | american | sales | home | large | hour | member |
| 10: | com | games | law | cost | won | food | small | nation |

Table 3: BOW:NYTIMES dataset [31]. The table lists the words corresponding to the $s = 10$ nonzero entries of each of the $k = 8$ extracted components (topics). Words corresponding to higher magnitude entries appear higher in the topic.

[4] R. Jiang, H. Fei, and J. Huan, "Anomaly localization for network data streams with graph joint sparse pca," in *Proceedings of the 17th ACM SIGKDD*, pp. 886–894, ACM, 2011.

[5] H. Zou, T. Hastie, and R. Tibshirani, "Sparse principal component analysis," *Journal of computational and graphical statistics*, vol. 15, no. 2, pp. 265–286, 2006.

[6] H. Kaiser, "The varimax criterion for analytic rotation in factor analysis," *Psychometrika*, vol. 23, no. 3, pp. 187–200, 1958.

[7] I. Jolliffe, "Rotation of principal components: choice of normalization constraints," *Journal of Applied Statistics*, vol. 22, no. 1, pp. 29–35, 1995.

[8] I. Jolliffe, N. Trendafilov, and M. Uddin, "A modified principal component technique based on the lasso," *Journal of Computational and Graphical Statistics*, vol. 12, no. 3, pp. 531–547, 2003.

[9] C. Boutsidis, P. Drineas, and M. Magdon-Ismail, "Sparse features for pca-like linear regression," in *Advances in Neural Information Processing Systems*, pp. 2285–2293, 2011.

[10] M. Journée, Y. Nesterov, P. Richtárik, and R. Sepulchre, "Generalized power method for sparse principal component analysis," *The Journal of Machine Learning Research*, vol. 11, pp. 517–553, 2010.

[11] B. Moghaddam, Y. Weiss, and S. Avidan, "Spectral bounds for sparse pca: Exact and greedy algorithms," *NIPS*, vol. 18, p. 915, 2006.

[12] A. d'Aspremont, F. Bach, and L. E. Ghaoui, "Optimal solutions for sparse principal component analysis," *The Journal of Machine Learning Research*, vol. 9, pp. 1269–1294, 2008.

[13] Y. Zhang, A. d'Aspremont, and L. Ghaoui, "Sparse pca: Convex relaxations, algorithms and applications," *Handbook on Semidefinite, Conic and Polynomial Optimization*, pp. 915–940, 2012.

[14] X.-T. Yuan and T. Zhang, "Truncated power method for sparse eigenvalue problems," *The Journal of Machine Learning Research*, vol. 14, no. 1, pp. 899–925, 2013.

[15] C. D. Sigg and J. M. Buhmann, "Expectation-maximization for sparse and non-negative pca," in *Proceedings of the 25th International Conference on Machine Learning*, ICML '08, (New York, NY, USA), pp. 960–967, ACM, 2008.

[16] D. Papailiopoulos, A. Dimakis, and S. Korokythakis, "Sparse pca through low-rank approximations," in *Proceedings of The 30th International Conference on Machine Learning*, pp. 747–755, 2013.

[17] M. Asteris, D. S. Papailiopoulos, and G. N. Karystinos, "The sparse principal component of a constant-rank matrix," *Information Theory, IEEE Transactions on*, vol. 60, pp. 2281–2290, April 2014.

[18] L. Mackey, "Deflation methods for sparse pca," *NIPS*, vol. 21, pp. 1017–1024, 2009.

[19] V. Vu and J. Lei, "Minimax rates of estimation for sparse pca in high dimensions," in *International Conference on Artificial Intelligence and Statistics*, pp. 1278–1286, 2012.

[20] M. Magdon-Ismail and C. Boutsidis, "Optimal sparse linear auto-encoders and sparse pca," *arXiv preprint arXiv:1502.06626*, 2015.

[21] M. Magdon-Ismail, "Np-hardness and inapproximability of sparse PCA," *CoRR*, vol. abs/1502.05675, 2015.

[22] L. Ramshaw and R. E. Tarjan, "On minimum-cost assignments in unbalanced bipartite graphs," *HP Labs, Palo Alto, CA, USA, Tech. Rep. HPL-2012-40R1*, 2012.

[23] N. Halko, P.-G. Martinsson, and J. A. Tropp, "Finding structure with randomness: Probabilistic algorithms for constructing approximate matrix decompositions," *SIAM review*, vol. 53, no. 2, pp. 217–288, 2011.

[24] M. Asteris, D. Papailiopoulos, A. Kyrillidis, and A. G. Dimakis, "Sparse pca via bipartite matchings," *arXiv preprint arXiv:1508.00625*, 2015.

[25] I. M. Johnstone and A. Y. Lu, "On consistency and sparsity for principal components analysis in high dimensions," *Journal of the American Statistical Association*, vol. 104, no. 486, 2009.

[26] Z. Ma, "Sparse principal component analysis and iterative thresholding," *The Annals of Statistics*, vol. 41, no. 2, pp. 772–801, 2013.

[27] V. Q. Vu, J. Cho, J. Lei, and K. Rohe, "Fantope projection and selection: A near-optimal convex relaxation of sparse pca," in *NIPS*, pp. 2670–2678, 2013.

[28] Z. Wang, H. Lu, and H. Liu, "Nonconvex statistical optimization: minimax-optimal sparse pca in polynomial time," *arXiv preprint arXiv:1408.5352*, 2014.

[29] R. Jenatton, G. Obozinski, and F. Bach, "Structured sparse principal component analysis," in *Proceedings of the Thirteenth International Conference on Artificial Intelligence and Statistics, AISTATS*, pp. 366–373, 2010.

[30] E. Richard, G. R. Obozinski, and J.-P. Vert, "Tight convex relaxations for sparse matrix factorization," in *Advances in Neural Information Processing Systems*, pp. 3284–3292, 2014.

[31] M. Lichman, "UCI machine learning repository," 2013.

