[Supplementary Material]

## Supplemental Material

## 7   On the sub-optimality of deflation – An example

We provide a simple example demonstrating the sub-optimality of deflation based approaches for computing multiple sparse components with disjoint supports. Consider the real $4 \times 4$ matrix

$$\mathbf{A} = \begin{bmatrix} 1 & 0 & 0 & \epsilon \\ 0 & \delta & 0 & 0 \\ 0 & 0 & \delta & 0 \\ \epsilon & 0 & 0 & 1 \end{bmatrix},$$

with $\epsilon, \delta > 0$ such that $\epsilon + \delta < 1$. Note that $\mathbf{A}$ is PSD; $\mathbf{A} = \mathbf{B}^\top \mathbf{B}$ for

$$\mathbf{B} = \begin{bmatrix} 1 & 0 & 0 & \epsilon \\ 0 & \sqrt{\delta} & 0 & 0 \\ 0 & 0 & \sqrt{\delta} & 0 \\ 0 & 0 & 0 & \sqrt{1 - \epsilon^2} \end{bmatrix}.$$

We seek two 2-sparse components with disjoint supports, *i.e.*, the solution to

$$\max_{\mathbf{X} \in \mathcal{X}} \sum_{j=1}^{2} \mathbf{x}_j^\top \mathbf{A} \mathbf{x}_j, \tag{8}$$

where

$$\mathcal{X} \triangleq \left\{ \mathbf{X} \in \mathbb{R}^{4 \times 2} : \|\mathbf{x}_i\|_2 \le 1, \|\mathbf{x}_i\|_0 \le 2 \; \forall \, i \in \{1, 2\}, \text{supp}(\mathbf{x}_1) \cap \text{supp}(\mathbf{x}_2) = \emptyset \right\}.$$

**Iterative computation with deflation.**   Following an iterative, greedy procedure with a deflation step, we compute one component at the time. The first component is

$$\mathbf{x}_1 = \arg\max_{\|\mathbf{x}\|_0 = 2, \|\mathbf{x}\|_2 = 1} \mathbf{x}^\top \mathbf{A} \mathbf{x}. \tag{9}$$

Recall that for any unit norm vector $\mathbf{x}$ with support $I = \text{supp}(\mathbf{x})$,

$$\mathbf{x}^\top \mathbf{A} \mathbf{x} \le \lambda_{\max} \left( \mathbf{A}_{I,I} \right), \tag{10}$$

where $\mathbf{A}_{I,I}$ denotes the principal submatrix of $\mathbf{A}$ formed by the rows and columns indexed by $I$. Equality can be achieved in (10) for $\mathbf{x}$ equal to the leading eigenvector of $\mathbf{A}_{I,I}$. Hence, it suffices to determine the optimal support for $\mathbf{x}_1$. Due to the small size of the example, it is easy to determine that the set $I_1 = \{1, 4\}$ maximizes the objective in (10) over all sets of two indices, achieving value

$$\mathbf{x}_1^\top \mathbf{A} \mathbf{x}_1 = \lambda_{\max} \left( \begin{bmatrix} 1 & \epsilon \\ \epsilon & 1 \end{bmatrix} \right) = 1 + \epsilon. \tag{11}$$

Since subsequent components must have disjoint supports, it follows that the support of the second 2-sparse component $\mathbf{x}_2$ is $I_2 = \{2, 3\}$, and $\mathbf{x}_2$ achieves value

$$\mathbf{x}_2^\top \mathbf{A} \mathbf{x}_2 = \lambda_{\max} \left( \begin{bmatrix} \delta & 0 \\ 0 & \delta \end{bmatrix} \right) = \delta. \tag{12}$$

In total, the objective value in (8) achieved by the greedy computation with a deflation step is

$$\sum_{j=1}^{2} \mathbf{x}_j^\top \mathbf{A} \mathbf{x}_j = 1 + \epsilon + \delta. \tag{13}$$

**The sub-optimality of deflation.**   Consider an alternative pair of 2-sparse components $\mathbf{x}_1'$ and $\mathbf{x}_2'$ with support sets $I_1' = \{1, 2\}$ and $I_2' = \{3, 4\}$, respectively. Based on the above, such a pair achieves objective value in (8) equal to

$$\lambda_{\max} \left( \begin{bmatrix} 1 & 0 \\ 0 & \delta \end{bmatrix} \right) + \lambda_{\max} \left( \begin{bmatrix} \delta & 0 \\ 0 & 1 \end{bmatrix} \right) = 1 + 1 = 2,$$

which clearly outperforms the objective value in (13) (under the assumption $\epsilon + \delta < 1$), demonstrating the sub-optimality of the $\mathbf{x}_1$, $\mathbf{x}_2$ pair computed by the deflation-based approach. In fact, for small $\epsilon, \delta$ the objective value in the second case is larger than the former by almost a factor of two.

# 8 Construction of Bipartite Graph

The following algorithm formally outlines the steps for generating the bipartite graph $G = \left(\{U_j\}_{j=1}^k, V, E\right)$ given a *weight* $d \times k$ matrix $\mathbf{W}$.

---

**Algorithm 4** Generate Bipartite Graph

---

**input** Real $d \times k$ matrix $\mathbf{W}$
**output** Bipartite $G = \left(\{U_j\}_{j=1}^k, V, E\right)$ {Fig. 1}
 1: **for** $j = 1, \ldots, k$ **do**
 2:     $U_j \leftarrow \left\{u_1^{(j)}, \ldots, u_s^{(j)}\right\}$
 3: **end for**
 4: $U \leftarrow \cup_{j=1}^k U_j$                                                         {$|U| = k \cdot s$}
 5: $V \leftarrow \{1, \ldots, d\}$
 6: $E \leftarrow U \times V$
 7: **for** $i = 1, \ldots, d$ **do**
 8:     **for** $j = 1, \ldots, k$ **do**
 9:         **for each** $u \in U_j$ **do**
10:            $w(u, v_i) \leftarrow W_{ij}^2$
11:         **end for**
12:     **end for**
13: **end for**

---

# 9 Proofs

## 9.1 Guarantees of Algorithm 2

**Lemma 2.1.** *For any real $d \times k$ matrix $\mathbf{W}$, and Algorithm 2 outputs*

$$\widetilde{\mathbf{X}} = \underset{\mathbf{X} \in \mathcal{X}_k}{\arg\max} \sum_{j=1}^k \left\langle \mathbf{X}^j, \mathbf{W}^j \right\rangle^2 \tag{14}$$

*in time $O\left(d \cdot (s \cdot k)^2\right)$.*

*Proof.* Consider a matrix $\mathbf{X} \in \mathcal{X}_k$ and let $I_j, j = 1 \ldots, k$ denote the support sets of its columns. By the constraints in $\mathcal{X}_k$, those sets are disjoint, *i.e.*, $I_{j_1} \cap I_{j_2} = \emptyset \ \forall j_1, j_2 \in \{1, \ldots, k\}, j_1 \neq j_2$, and

$$\sum_{j=1}^k \left\langle \mathbf{X}^j, \mathbf{W}^j \right\rangle^2 = \sum_{j=1}^k \left( \sum_{i \in I_j} X_{ij} \cdot W_{ij} \right)^2 \leq \sum_{j=1}^k \left( \sum_{i \in I_j} W_{ij}^2 \right). \tag{15}$$

The last inequality is due to Cauchy-Schwarz and the fact that $\|\mathbf{X}^j\|_2 \leq 1, \forall j \in \{1, \ldots, k\}$. In fact, if the supports sets $I_j, j = 1, \ldots, k$ were known, the upper bound in (15) would be achieved by setting $\mathbf{X}_{I_j}^j = \mathbf{W}_{I_j}^j / \|\mathbf{W}_{I_j}^j\|_2$, *i.e.*, setting the nonzero subvector of the $j$th column of $\mathbf{X}$ colinear to the corresponding subvector of the $j$th column of $\mathbf{W}$. Hence, the key step towards computing the optimal solution $\widetilde{\mathbf{X}}$ is to determine the support sets $I_j, j = 1, \ldots, k$ of its columns.

Consider the set of binary matrices

$$\mathcal{Z} \triangleq \left\{ \mathbf{Z} \in \{0,1\}^{d \times k} : \|\mathbf{Z}^j\|_0 \leq s \ \forall j \in [k], \mathrm{supp}(\mathbf{Z}^i) \cap \mathrm{supp}(\mathbf{Z}^j) = \emptyset \ \forall i, j \in [k], i \neq j \right\}.$$

The set represents all possible supports for the members of $\mathcal{X}_k$. Taking into account the previous discussion, the maximization in (14) can be written with respect to $\mathbf{Z} \in \mathcal{Z}$:

$$\max_{\mathbf{X} \in \mathcal{X}_k} \sum_{j=1}^k \left\langle \mathbf{X}^j, \mathbf{W}^j \right\rangle^2 = \max_{\mathbf{Z} \in \mathcal{Z}} \sum_{j=1}^k \sum_{i=1}^d Z_{ij} W_{ij}^2. \tag{16}$$

Let $\widetilde{\mathbf{Z}} \in \mathcal{Z}$ denote the optimal solution, which corresponds to the (support) indicator of $\widetilde{\mathbf{X}}$. Next, we show that computing $\widetilde{\mathbf{Z}}$ boils down to solving a maximum weight matching problem on the bipartite graph generated by Algorithm 4. Recall that given $\mathbf{W} \in \mathbb{R}^{d \times k}$, Algorithm 4 generates a complete weighted bipartite graph $G = (U, V, E)$ where

- $V$ is a set of $d$ vertices $v_1, \ldots, v_d$, corresponding to the $d$ variables, *i.e.*, the $d$ rows of $\widehat{\mathbf{X}}$.
- $U$ is a set of $k \cdot s$ vertices, conceptually partitioned into $k$ disjoint subsets $U_1, \ldots, U_k$, each of cardinality $s$. The $j$th subset, $U_j$, is associated with the support $\mathcal{I}_j$; the $s$ vertices $u_\alpha^{(j)}, \alpha = 1, \ldots, s$ in $U_j$ serve as placeholders for the variables/indices in $\mathcal{I}_j$.

- Finally, the edge set is $E = U \times V$. The edge weights are determined by the $d \times k$ matrix $\mathbf{W}$ in (6). In particular, the weight of edge $(u_\alpha^{(j)}, v_i)$ is equal to $W_{ij}^2$. Note that all vertices in $U_j$ are effectively identical; they all share a common neighborhood and edge weights.

It is straightforward to verify that any $\mathbf{Z} \in \mathcal{Z}$ corresponds to a perfect matching in $G$ and vice versa; $Z_{ij} = 1$ if and only if vertex $v_i \in V$ is matched with a vertex in $U_j$ (all vertices in $U_j$ are equivalent with respect to their neighborhood). Further, for a given $\mathbf{Z} \in \mathcal{Z}$ the objective value in (16) is equal to the weight of the corresponding matching in $G$. More formally, For a given perfect matching $\mathcal{M} \subset E$, the corresponding indicator matrix $\mathbf{Z} \in \mathcal{Z}$ (and equivalently the support of its columns) is determined by setting

$$I_j \leftarrow \big\{ i \in [d] : (u, v_i) \in \mathcal{M}, u \in U_j \big\}, \quad j = 1, \dots, k. \tag{17}$$

The weight of the matching $\mathcal{M}$ is

$$\sum_{(u,v)\in\mathcal{M}} w(u,v) = \sum_{j=1}^{k} \sum_{\substack{(u,v_i)\in\mathcal{M}: \\ u \in U_j}} w(u, v_i) = \sum_{j=1}^{k} \sum_{i \in I_j} W_{ij}^2 = \sum_{j=1}^{k} \sum_{i=1}^{d} Z_{ij} \cdot W_{ij}^2, \tag{18}$$

which is equal to the objective function in (16). Conversely, any given indicator matrix $\mathbf{Z} \in \mathcal{Z}$ corresponds to a perfect matching $\mathcal{M} \subset E$. In particular, letting $I_j \triangleq \mathrm{supp}(\mathbf{Z}^j)$, and for an arbitrary ordering $\sigma_j : [s] \to I_j$ of the elements of $I_j$,

$$\mathcal{M} \leftarrow \Big\{ (u_\alpha^{(j)}, v_{\sigma_j(\alpha)}), \alpha = 1, \dots, s, j = 1, \dots, k \Big\}$$

is a perfect matching in $G$. The weight of the matching $\mathcal{M}$ is equal to the objective value in (16) for that $\mathbf{Z}$:

$$\sum_{j=1}^{k} \sum_{i=1}^{d} Z_{ij} \cdot W_{ij}^2 = \sum_{j=1}^{k} \sum_{i \in I_j} W_{ij}^2 = \sum_{j=1}^{k} \sum_{\alpha=1}^{s} W_{I_j(\alpha),j}^2 = \sum_{(u,v)\in\mathcal{M}} w(u,v). \tag{19}$$

It follows that to determine $\widetilde{\mathbf{Z}}$ that maximizes (16) with respect to $\mathbf{Z} \in \mathcal{Z}$, it suffices to compute a maximum weight perfect matching in $G$. Then $\widetilde{\mathbf{Z}}$ is obtained as described in (17). Finally, the values of the non-zero entries of $\widetilde{\mathbf{X}}$ are determined as described in the beginning of the proof (lines 4-7 of Algorithm 2), guaranteeing the optimality of $\widetilde{\mathbf{X}}$ for the maximization in (14).

The weighted bipartite graph $G$ is generated in $O(d \cdot (s \cdot k))$. The running time of Algorithm 2 is dominated by the computation of the maximum weight matching of $G$. For the case of unbalanced bipartite graph with $|U| = s \cdot k < d = |V|$ the Hungarian algorithm can be modified [22] to compute the maximum weight bipartite matching in time $O\big(|E||U| + |U|^2 \log |U|\big) = O\big(d \cdot (s \cdot k)^2\big)$. This completes the proof. $\qquad\square$

## 9.2 Guarantees of Algorithm 1 – Proof of Theorem 1

We first prove a more general version of Theorem 1 for arbitrary constraint sets. Combining that with the guarantees of Algorithm 2, we prove the Theorem 1.

**Lemma 9.2.** *For any real $d \times d$ rank-$r$ PSD matrix $\overline{\mathbf{A}}$ and arbitrary set $\mathcal{X} \subset \mathbb{R}^{d \times k}$, let $\overline{\mathbf{X}}_\star \triangleq \arg\max_{\mathbf{X} \in \mathcal{X}} \mathrm{Tr}\big(\mathbf{X}^\top \overline{\mathbf{A}} \mathbf{X}\big)$. Assuming that there exists an operator $P_\mathcal{X} : \mathbb{R}^{d \times k} \to \mathcal{X}$ such that $P_\mathcal{X}(\mathbf{W}) = \arg\max_{\mathbf{X} \in \mathcal{X}} \langle \mathbf{x}_j, \mathbf{w}_j \rangle^2$, then Algorithm 1 outputs $\overline{\mathbf{X}} \in \mathcal{X}$ such that*

$$\mathrm{Tr}\big(\overline{\mathbf{X}}^\top \overline{\mathbf{A}} \overline{\mathbf{X}}\big) \geq (1 - \epsilon) \cdot \mathrm{Tr}\big(\overline{\mathbf{X}}_\star^\top \overline{\mathbf{A}} \overline{\mathbf{X}}_\star\big),$$

*in time $T_{SVD}(r) + O\big(\big(\frac{4}{\epsilon}\big)^{r \cdot k} \cdot (T_\mathcal{X} + kd)\big)$, where $T_\mathcal{X}$ is the time required to compute $P_\mathcal{X}(\cdot)$ and $T_{SVD}(r)$ the time required to compute the truncated SVD of $\overline{\mathbf{A}}$.*

*Proof.* Let $\overline{\mathbf{A}} = \overline{\mathbf{U}} \overline{\mathbf{\Lambda}} \overline{\mathbf{U}}^\top$ denote the truncated eigenvalue decomposition of $\overline{\mathbf{A}}$; $\overline{\mathbf{\Lambda}}$ is a diagonal $r \times r$ whose $i$th diagonal entry $\Lambda_{ii}$ is equal to the $i$th largest eigenvalue of $\overline{\mathbf{A}}$, while the columns of $\overline{\mathbf{U}}$ contain the corresponding eigenvectors. By the Cauchy-Schwartz inequality, for any $\mathbf{x} \in \mathbb{R}^d$,

$$\mathbf{x}^\top \overline{\mathbf{A}} \mathbf{x} = \big\| \overline{\mathbf{\Lambda}}^{1/2} \overline{\mathbf{U}}^\top \mathbf{x} \big\|_2^2 \geq \big\langle \overline{\mathbf{\Lambda}}^{1/2} \overline{\mathbf{U}}^\top \mathbf{x}, \mathbf{c} \big\rangle^2, \quad \forall \mathbf{c} \in \mathbb{R}^r : \|\mathbf{c}\|_2 = 1. \tag{20}$$

In fact, equality in (20) is achieved for $\mathbf{c}$ colinear to $\overline{\mathbf{\Lambda}}^{1/2} \overline{\mathbf{U}} \mathbf{x}$, and hence,

$$\mathbf{x}^\top \overline{\mathbf{A}} \mathbf{x} = \max_{\mathbf{c} \in \mathbb{S}_2^{r-1}} \big\langle \overline{\mathbf{\Lambda}}^{1/2} \overline{\mathbf{U}}^\top \mathbf{x}, \mathbf{c} \big\rangle^2. \tag{21}$$

In turn,

$$\mathrm{Tr}\Big(\mathbf{X}^\top \overline{\mathbf{A}} \mathbf{X}\Big) = \sum_{j=1}^{k} \mathbf{X}^{j^\top} \overline{\mathbf{A}} \mathbf{X}^j = \max_{\mathbf{C}:\mathbf{C}^j \in \mathbb{S}_2^{r-1} \forall j} \sum_{j=1}^{k} \big\langle \overline{\mathbf{\Lambda}}^{1/2} \overline{\mathbf{U}}^\top \mathbf{X}^j, \mathbf{C}^j \big\rangle^2. \tag{22}$$

Recall that $\overline{\mathbf{X}}_\star$ is the optimal solution of the trace maximization on $\overline{\mathbf{A}}$, *i.e.*,

$$\overline{\mathbf{X}}_\star \triangleq \arg\max_{\mathbf{X}\in\mathcal{X}} \mathrm{T}_R\left(\mathbf{X}^\top \overline{\mathbf{A}}\mathbf{X}\right).$$

Let $\overline{\mathbf{C}}_\star$ be the maximizing value of $\mathbf{C}$ in (22) for $\mathbf{X} = \overline{\mathbf{X}}_\star$, *i.e.*, $\overline{\mathbf{C}}_\star$ is an $r \times k$ matrix with unit-norm columns such that for all $j \in \{1,\ldots,k\}$,

$$\overline{\mathbf{X}}_\star^{j\top}\overline{\mathbf{A}}\overline{\mathbf{X}}_\star^j = \langle \overline{\mathbf{\Lambda}}^{1/2}\overline{\mathbf{U}}^\top \overline{\mathbf{X}}_\star^j, \overline{\mathbf{C}}_\star^j\rangle^2. \tag{23}$$

Algorithm 1 iterates over the points ($r \times k$ matrices) $\mathbf{C}$ in $\mathcal{N}_{\epsilon/2}^{\otimes k}\left(\mathbb{S}_2^{r-1}\right)$, the $k$th cartesian power of a finite $\epsilon/2$-net of the $r$-dimensional $l_2$-unit sphere. At each such point $\mathbf{C}$, it computes a candidate

$$\widetilde{\mathbf{X}} = \arg\max_{\mathbf{X}\in\mathcal{X}}\sum_{j=1}^k \langle \mathbf{X}^j, \mathbf{U}\mathbf{\Lambda}^{1/2}\mathbf{C}^j\rangle^2$$

via Algorithm 2 (See Lemma 9.1 for the guarantees of Algorithm 2). By construction, the set $\mathcal{N}_{\epsilon/2}^{\otimes k}\left(\mathbb{S}_2^{r-1}\right)$ contains a $\mathbf{C}_\sharp$ such that

$$\|\mathbf{C}_\sharp - \overline{\mathbf{C}}_\star\|_{\infty,2} = \max_{j\in\{1,\ldots,k\}}\|\mathbf{C}_\sharp^j - \overline{\mathbf{C}}_\star^j\|_2 \le \epsilon/2. \tag{24}$$

Based on the above, for all $j \in \{1,\ldots,k\}$,

$$\begin{aligned}
(\overline{\mathbf{X}}_\star^{j\top}\overline{\mathbf{A}}\overline{\mathbf{X}}_\star^j)^{1/2} &= |\langle \overline{\mathbf{\Lambda}}^{1/2}\overline{\mathbf{U}}^\top \overline{\mathbf{X}}_\star^j, \overline{\mathbf{C}}_\star^j\rangle|\\
&= |\langle \overline{\mathbf{\Lambda}}^{1/2}\overline{\mathbf{U}}^\top \overline{\mathbf{X}}_\star^j, \mathbf{C}_\sharp^j\rangle + \langle \overline{\mathbf{\Lambda}}^{1/2}\overline{\mathbf{U}}^\top \overline{\mathbf{X}}_\star^j, (\overline{\mathbf{C}}_\star^j - \mathbf{C}_\sharp^j)\rangle|\\
&\le |\langle \overline{\mathbf{\Lambda}}^{1/2}\overline{\mathbf{U}}^\top \overline{\mathbf{X}}_\star^j, \mathbf{C}_\sharp^j\rangle| + |\langle \overline{\mathbf{\Lambda}}^{1/2}\overline{\mathbf{U}}^\top \overline{\mathbf{X}}_\star^j, (\overline{\mathbf{C}}_\star^j - \mathbf{C}_\sharp^j)\rangle|\\
&\le |\langle \overline{\mathbf{\Lambda}}^{1/2}\overline{\mathbf{U}}^\top \overline{\mathbf{X}}_\star^j, \mathbf{C}_\sharp^j\rangle| + \|\overline{\mathbf{\Lambda}}^{1/2}\overline{\mathbf{U}}^\top \overline{\mathbf{X}}_\star^j\| \cdot \|\overline{\mathbf{C}}_\star^j - \mathbf{C}_\sharp^j\|\\
&\le |\langle \overline{\mathbf{\Lambda}}^{1/2}\overline{\mathbf{U}}^\top \overline{\mathbf{X}}_\star^j, \mathbf{C}_\sharp^j\rangle| + (\epsilon/2)\cdot(\overline{\mathbf{X}}_\star^{j\top}\overline{\mathbf{A}}\overline{\mathbf{X}}_\star^j)^{1/2}. 
\end{aligned} \tag{25}$$

The first step follows by the definition of $\overline{\mathbf{C}}_\star$, the second by the linearity of the inner product, the third by the triangle inequality, the fourth by Cauchy-Schwarz inequality and the last by (24). Rearranging the terms in (25),

$$|\langle \overline{\mathbf{\Lambda}}^{1/2}\overline{\mathbf{U}}^\top \overline{\mathbf{X}}_\star^j, \mathbf{C}_\sharp^j\rangle| \ge \left(1 - \tfrac{\epsilon}{2}\right)\cdot(\overline{\mathbf{X}}_\star^{j\top}\overline{\mathbf{A}}\overline{\mathbf{X}}_\star^j)^{1/2} \ge 0,$$

and in turn,

$$\langle \overline{\mathbf{\Lambda}}^{1/2}\overline{\mathbf{U}}^\top \overline{\mathbf{X}}_\star^j, \mathbf{C}_\sharp^j\rangle^2 \ge \left(1 - \tfrac{\epsilon}{2}\right)^2 \cdot \overline{\mathbf{X}}_\star^{j\top}\overline{\mathbf{A}}\overline{\mathbf{X}}_\star^j \ge (1 - \epsilon)\cdot\overline{\mathbf{X}}_\star^{j\top}\overline{\mathbf{A}}\overline{\mathbf{X}}_\star^j \tag{26}$$

Summing the terms in (26) over all $j \in \{1,\ldots,k\}$,

$$\sum_{j=1}^k \langle \overline{\mathbf{\Lambda}}^{1/2}\overline{\mathbf{U}}^\top \overline{\mathbf{X}}_\star^j, \mathbf{C}_\sharp^j\rangle^2 \ge (1 - \epsilon)\cdot\mathrm{T}_R\left(\overline{\mathbf{X}}_\star^\top \overline{\mathbf{A}}\overline{\mathbf{X}}_\star\right). \tag{27}$$

Let $\mathbf{X}_\sharp \in \mathcal{X}$ be the candidate solution produced by the algorithm at $\mathbf{C}_\sharp$, *i.e.*,

$$\mathbf{X}_\sharp \triangleq \arg\max_{\mathbf{X}\in\mathcal{X}}\sum_{j=1}^k \langle \mathbf{x}_j, \overline{\mathbf{U}}\overline{\mathbf{\Lambda}}^{1/2}\mathbf{C}_\sharp^j\rangle^2. \tag{28}$$

Then,

$$\begin{aligned}
\mathrm{T}_R\left(\mathbf{X}_\sharp^\top \overline{\mathbf{A}}\mathbf{X}_\sharp\right) &\stackrel{(\alpha)}{=} \max_{\mathbf{C}:\mathbf{C}^j\in\mathbb{S}_2^{r-1}\,\forall j}\sum_{j=1}^k \langle \overline{\mathbf{\Lambda}}^{1/2}\overline{\mathbf{U}}^\top \overline{\mathbf{X}}_\sharp^j, \mathbf{C}^j\rangle^2\\
&\stackrel{(\beta)}{\ge} \sum_{j=1}^k \langle \overline{\mathbf{\Lambda}}^{1/2}\overline{\mathbf{U}}^\top \overline{\mathbf{X}}_\sharp^j, \mathbf{C}_\sharp^j\rangle^2\\
&\stackrel{(\gamma)}{\ge} \sum_{j=1}^k \langle \overline{\mathbf{X}}_\star^j, \overline{\mathbf{U}}\overline{\mathbf{\Lambda}}^{1/2}\mathbf{C}_\sharp^j\rangle^2\\
&\stackrel{(\delta)}{\ge} (1 - \epsilon)\cdot\mathrm{T}_R\left(\overline{\mathbf{X}}_\star^\top \overline{\mathbf{A}}\overline{\mathbf{X}}_\star\right),
\end{aligned} \tag{29}$$

where $(\alpha)$ follows from the observation in (22), $(\beta)$ from the sub-optimality of $\mathbf{C}_\sharp$, $(\gamma)$ by the definition of $\mathbf{X}_\sharp$ in (28), while $(\delta)$ follows from (27). According to (29), at least one of the candidate solutions produced by Algorithm 1, namely $\mathbf{X}_\sharp$, achieves an objective value within a multiplicative factor $(1 - \epsilon)$ from the optimal, implying the guarantees of the lemma.

Finally, the running time of Algorithm 1 follows immediately from the cost per iteration and the cardinality of the $\epsilon/2$-net on the unit-sphere. Note that matrix multiplications can exploit the singular value decomposition which is performed once. $\qquad\square$

**Theorem 1.** *For any real $d \times d$ rank-r PSD matrix $\overline{\mathbf{A}}$, desired number of components $k$, number $s$ of nonzero entries per component, and accuracy parameter $\epsilon \in (0,1)$, Algorithm 1 outputs $\overline{\mathbf{X}} \in \mathcal{X}_k$ such that*

$$\mathrm{TR}\big(\overline{\mathbf{X}}^\top \overline{\mathbf{A}}\,\overline{\mathbf{X}}\big) \;\geq\; (1-\epsilon) \cdot \mathrm{TR}\big(\mathbf{X}_\star^\top \overline{\mathbf{A}}\mathbf{X}_\star\big),$$

*where $\mathbf{X}_\star \triangleq \arg\max_{\mathbf{X} \in \mathcal{X}_k} \mathrm{TR}\big(\mathbf{X}^\top \overline{\mathbf{A}}\mathbf{X}\big)$, in time $T_{SVD}(r) + O\big(\big(\tfrac{4}{\epsilon}\big)^{r \cdot k} \cdot d \cdot (s \cdot k)^2\big)$. $T_{SVD}(r)$ is the time required to compute the truncated SVD of $\overline{\mathbf{A}}$.*

*Proof.* Recall that $\mathcal{X}_k$ is the set of $d \times k$ matrices $\mathbf{X}$ whose columns have unit length and pairwise disjoint supports. Algorithm 2, given any $\mathbf{W} \in \mathbb{R}^{d \times k}$, computes $\mathbf{X} \in \mathcal{X}_k$ that optimally solves the constrained maximization in line 5. (See Lemma 9.1 for the guarantee of Algorithm 2). in time $O\big(d \cdot (s \cdot k)^2\big)$. The desired result then follows by Lemma 9.2 for the constrained set $\mathcal{X}_k$. □

## 9.3 Guarantees of Algorithm 3 – Proof of Theorem 2

We prove Theorem 2 with the approximation guarantees of Algorithm 3.

**Lemma 9.3.** *For any $d \times d$ PSD matrices $\mathbf{A}$ and $\overline{\mathbf{A}}$, and any set $\mathcal{X} \subseteq \mathbb{R}^{d \times k}$ let*

$$\mathbf{X}_\star \triangleq \arg\max_{\mathbf{X} \in \mathcal{X}} \mathrm{TR}\big(\mathbf{X}^\top \mathbf{A}\mathbf{X}\big), \quad and \quad \overline{\mathbf{X}}_\star \triangleq \arg\max_{\mathbf{X} \in \mathcal{X}} \mathrm{TR}\big(\mathbf{X}^\top \overline{\mathbf{A}}\mathbf{X}\big).$$

*Then, for any $\overline{\mathbf{X}} \in \mathcal{X}$ such that $\mathrm{TR}\big(\overline{\mathbf{X}}^\top \overline{\mathbf{A}}\,\overline{\mathbf{X}}\big) \geq \gamma \cdot \mathrm{TR}\big(\overline{\mathbf{X}}_\star^\top \overline{\mathbf{A}}\overline{\mathbf{X}}_\star\big)$ for some $0 < \gamma < 1$,*

$$\mathrm{TR}\big(\overline{\mathbf{X}}^\top \mathbf{A}\overline{\mathbf{X}}\big) \geq \gamma \cdot \mathrm{TR}\big(\mathbf{X}_\star^\top \mathbf{A}\mathbf{X}_\star\big) - 2 \cdot \|\mathbf{A} - \overline{\mathbf{A}}\|_2 \cdot \max_{\mathbf{X} \in \mathcal{X}} \|\mathbf{X}\|_F^2.$$

*Proof.* By the optimality of $\overline{\mathbf{X}}_\star$ for $\overline{\mathbf{A}}$,

$$\mathrm{TR}\big(\overline{\mathbf{X}}_\star^\top \overline{\mathbf{A}}\overline{\mathbf{X}}_\star\big) \geq \mathrm{TR}\big(\mathbf{X}_\star^\top \overline{\mathbf{A}}\mathbf{X}_\star\big).$$

In turn, for any $\overline{\mathbf{X}} \in \mathcal{X}$ such that $\mathrm{TR}\big(\overline{\mathbf{X}}^\top \overline{\mathbf{A}}\overline{\mathbf{X}}\big) \geq \gamma \cdot \mathrm{TR}\big(\overline{\mathbf{X}}_\star^\top \overline{\mathbf{A}}\overline{\mathbf{X}}_\star\big)$ for some $0 < \gamma < 1$,

$$\mathrm{TR}\big(\overline{\mathbf{X}}^\top \overline{\mathbf{A}}\overline{\mathbf{X}}\big) \geq \gamma \cdot \mathrm{TR}\big(\mathbf{X}_\star^\top \overline{\mathbf{A}}\mathbf{X}_\star\big). \tag{30}$$

Let $\mathbf{E} \triangleq \mathbf{A} - \overline{\mathbf{A}}$. By the linearity of the trace,

$$\mathrm{TR}\big(\overline{\mathbf{X}}^\top \overline{\mathbf{A}}\overline{\mathbf{X}}\big) = \mathrm{TR}\big(\overline{\mathbf{X}}^\top \mathbf{A}\overline{\mathbf{X}}\big) - \mathrm{TR}\big(\overline{\mathbf{X}}^\top \mathbf{E}\overline{\mathbf{X}}\big)$$

$$\leq \mathrm{TR}\big(\overline{\mathbf{X}}^\top \mathbf{A}\overline{\mathbf{X}}\big) + \big|\mathrm{TR}\big(\overline{\mathbf{X}}^\top \mathbf{E}\overline{\mathbf{X}}\big)\big|. \tag{31}$$

By Lemma 10.9,

$$\big|\mathrm{TR}\big(\overline{\mathbf{X}}^\top \mathbf{E}\overline{\mathbf{X}}\big)\big| \leq \|\overline{\mathbf{X}}\|_F \cdot \|\overline{\mathbf{X}}\|_F \cdot \|\mathbf{E}\|_2 \leq \|\mathbf{E}\|_2 \cdot \max_{\mathbf{X} \in \mathcal{X}} \|\mathbf{X}\|_F^2 \triangleq R. \tag{32}$$

Continuing from (31),

$$\mathrm{TR}\big(\overline{\mathbf{X}}^\top \overline{\mathbf{A}}\overline{\mathbf{X}}\big) \leq \mathrm{TR}\big(\overline{\mathbf{X}}^\top \mathbf{A}\overline{\mathbf{X}}\big) + R. \tag{33}$$

Similarly,

$$\mathrm{TR}\big(\mathbf{X}_\star^\top \overline{\mathbf{A}}\mathbf{X}_\star\big) = \mathrm{TR}\big(\mathbf{X}_\star^\top \mathbf{A}\mathbf{X}_\star\big) - \mathrm{TR}\big(\mathbf{X}_\star^\top \mathbf{E}\mathbf{X}_\star\big)$$

$$\geq \mathrm{TR}\big(\mathbf{X}_\star^\top \mathbf{A}\mathbf{X}_\star\big) - \big|\mathrm{TR}\big(\mathbf{X}_\star^\top \mathbf{E}\mathbf{X}_\star\big)\big|$$

$$\geq \mathrm{TR}\big(\mathbf{X}_\star^\top \mathbf{A}\mathbf{X}_\star\big) - R. \tag{34}$$

Combining the above, we have

$$\mathrm{TR}\big(\overline{\mathbf{X}}^\top \mathbf{A}\overline{\mathbf{X}}\big) \geq \mathrm{TR}\big(\overline{\mathbf{X}}^\top \overline{\mathbf{A}}\overline{\mathbf{X}}\big) - R$$

$$\geq \gamma \cdot \mathrm{TR}\big(\mathbf{X}_\star^\top \overline{\mathbf{A}}\mathbf{X}_\star\big) - R$$

$$\geq \gamma \cdot \Big(\mathrm{TR}\big(\mathbf{X}_\star^\top \mathbf{A}\mathbf{X}_\star\big) - R\Big) - R$$

$$= \gamma \cdot \mathrm{TR}\big(\mathbf{X}_\star^\top \mathbf{A}\mathbf{X}_\star\big) - (1+\gamma) \cdot R$$

$$\geq \gamma \cdot \mathrm{TR}\big(\mathbf{X}_\star^\top \mathbf{A}\mathbf{X}_\star\big) - 2 \cdot R,$$

where the first inequality follows from (33) the second from (30), the third from (34), and the last from the fact that $R \geq 0$ and $0 < \gamma \leq 1$. This concludes the proof. □

**Remark 9.1.** *If in Lemma 9.3 the PSD matrices $\mathbf{A}$ and $\overline{\mathbf{A}} \in \mathbb{R}^{d \times d}$ are such that $\mathbf{A} - \overline{\mathbf{A}}$ is also PSD, then the following tighter bound holds:*

$$\mathrm{TR}\big(\overline{\mathbf{X}}^\top \mathbf{A} \overline{\mathbf{X}}\big) \geq \gamma \cdot \mathrm{TR}\big(\mathbf{X}_\star^\top \mathbf{A} \mathbf{X}_\star\big) - \sum_{i=1}^{k} \lambda_i \big(\mathbf{A} - \overline{\mathbf{A}}\big).$$

*Proof.* This follows from the fact that if $\mathbf{E} \triangleq \mathbf{A} - \overline{\mathbf{A}}$ is PSD, then

$$\mathrm{TR}\left(\overline{\mathbf{X}}^\top \mathbf{E} \overline{\mathbf{X}}\right) = \sum_{j=1}^{d} \mathbf{x}_j^\top \mathbf{E} \mathbf{x}_j \geq 0,$$

and the bound in (31) can be improved to

$$\mathrm{TR}\left(\overline{\mathbf{X}}^\top \overline{\mathbf{A}} \overline{\mathbf{X}}\right) = \mathrm{TR}\left(\overline{\mathbf{X}}^\top \mathbf{A} \overline{\mathbf{X}}\right) - \mathrm{TR}\left(\overline{\mathbf{X}}^\top \mathbf{E} \overline{\mathbf{X}}\right) \leq \mathrm{TR}\left(\overline{\mathbf{X}}^\top \mathbf{A} \overline{\mathbf{X}}\right).$$

Further, by Lemma 10.10, the bound in (32) can be improved to

$$\mathrm{TR}\big(\overline{\mathbf{X}}^\top \mathbf{E} \overline{\mathbf{X}}\big) \leq \sum_{i=1}^{k} \lambda_i \big(\mathbf{E}\big) \triangleq R.$$

The rest of the proof follows as is. $\qquad \square$

**Theorem 2.** *For any $n \times d$ input data matrix $\mathbf{S}$, with corresponding empirical covariance matrix $\mathbf{A} = 1/n \cdot \mathbf{S}^\top \mathbf{S}$, any desired number of components $k$, and accuracy parameters $\epsilon \in (0,1)$ and $r$, Algorithm 3 outputs $\mathbf{X}_{(r)} \in \mathcal{X}_k$ such that*

$$\mathrm{TR}\big(\mathbf{X}_{(r)}^\top \mathbf{A} \mathbf{X}_{(r)}\big) \geq (1-\epsilon) \cdot \mathrm{TR}\big(\mathbf{X}_\star^\top \mathbf{A} \mathbf{X}_\star\big) - 2 \cdot k \cdot \|\mathbf{A} - \overline{\mathbf{A}}\|_2,$$

*where $\mathbf{X}_\star \triangleq \arg\max_{\mathbf{X} \in \mathcal{X}_k} \mathrm{TR}\big(\mathbf{X}^\top \mathbf{A} \mathbf{X}\big)$, in time $T_{SKETCH}(r) + T_{SVD}(r) + O\big(\big(\frac{4}{\epsilon}\big)^{r \cdot k} \cdot d \cdot (s \cdot k)^2\big)$.*

*Proof.* The theorem follows from Lemma 9.3 and the approximation guarantees of Algorithm 1. $\qquad \square$

# 10  Auxiliary Technical Lemmata

**Lemma 10.4.** *For any real $d \times n$ matrix $\mathbf{M}$, and any $r, k \leq \min\{d, n\}$,*

$$\sum_{i=r+1}^{r+k} \sigma_i(\mathbf{M}) \leq \frac{k}{\sqrt{r+k}} \cdot \|\mathbf{M}\|_\mathrm{F},$$

*where $\sigma_i(\mathbf{M})$ is the $i$th largest singular value of $\mathbf{M}$.*

*Proof.* By the Cauchy-Schwartz inequality,

$$\sum_{i=r+1}^{r+k} \sigma_i(\mathbf{M}) = \sum_{i=r+1}^{r+k} |\sigma_i(\mathbf{M})| \leq \left(\sum_{i=r+1}^{r+k} \sigma_i^2(\mathbf{M})\right)^{1/2} \cdot \|\mathbf{1}_k\|_2 = \sqrt{k} \cdot \left(\sum_{i=r+1}^{r+k} \sigma_i^2(\mathbf{M})\right)^{1/2}.$$

Note that $\sigma_{r+1}(\mathbf{M}), \ldots, \sigma_{r+k}(\mathbf{M})$ are the $k$ smallest among the $r + k$ largest singular values. Hence,

$$\sum_{i=r+1}^{r+k} \sigma_i^2(\mathbf{M}) \leq \frac{k}{r+k} \sum_{i=1}^{r+k} \sigma_i^2(\mathbf{M}) \leq \frac{k}{r+k} \sum_{i=1}^{\min\{d,n\}} \sigma_i^2(\mathbf{M}) = \frac{k}{r+k} \|\mathbf{M}\|_\mathrm{F}^2.$$

Combining the two inequalities, the desired result follows. $\qquad \square$

**Corollary 1.** *For any real $d \times n$ matrix $\mathbf{M}$ and $k \leq \min\{d, n\}$, $\sigma_k(\mathbf{M}) \leq k^{-1/2} \cdot \|\mathbf{M}\|_\mathrm{F}$.*

*Proof.* It follows immediately from Lemma 10.4. $\qquad \square$

**Lemma 10.5.** *Let $a_1, \ldots, a_n$ and $b_1, \ldots, b_n$ be $2n$ real numbers and let $p$ and $q$ be two numbers such that $1/p + 1/q = 1$ and $p > 1$. We have*

$$|\sum_{i=1}^{n} a_i b_i| \leq \left(\sum_{i=1}^{n} |a_i|^p\right)^{1/p} \cdot \left(\sum_{i=1}^{n} |b_i|^q\right)^{1/q}.$$

**Lemma 10.6.** *For any two real matrices $\mathbf{A}$ and $\mathbf{B}$ of appropriate dimensions,*

$$\|\mathbf{A}\mathbf{B}\|_\mathrm{F} \leq \min\{\|\mathbf{A}\|_2 \|\mathbf{B}\|_\mathrm{F}, \ \|\mathbf{A}\|_\mathrm{F} \|\mathbf{B}\|_2\}.$$

*Proof.* Let $\mathbf{b}_i$ denote the $i$th column of $\mathbf{B}$. Then,

$$\|\mathbf{AB}\|_{\mathrm{F}}^2 = \sum_i \|\mathbf{Ab}_i\|_2^2 \leq \sum_i \|\mathbf{A}\|_2^2 \|\mathbf{b}_i\|_2^2 = \|\mathbf{A}\|_2^2 \sum_i \|\mathbf{b}_i\|_2^2 = \|\mathbf{A}\|_2^2 \|\mathbf{B}\|_{\mathrm{F}}^2.$$

Similarly, using the previous inequality,

$$\|\mathbf{AB}\|_{\mathrm{F}}^2 = \|\mathbf{B}^\top \mathbf{A}^\top\|_{\mathrm{F}}^2 \leq \|\mathbf{B}^\top\|_2^2 \|\mathbf{A}^\top\|_{\mathrm{F}}^2 = \|\mathbf{B}\|_2^2 \|\mathbf{A}\|_{\mathrm{F}}^2.$$

Combining the two upper bounds, the desired result follows. $\quad\square$

**Lemma 10.7.** *For any* $\mathbf{A}, \mathbf{B} \in \mathbb{R}^{n \times k}$,

$$|\langle \mathbf{A}, \mathbf{B} \rangle| \triangleq \left|\mathrm{TR}\left(\mathbf{A}^\top \mathbf{B}\right)\right| \leq \|\mathbf{A}\|_{\mathrm{F}} \|\mathbf{B}\|_{\mathrm{F}}.$$

*Proof.* The inequality follows from Lemma 10.5 for $p = q = 2$, treating $\mathbf{A}$ and $\mathbf{B}$ as vectors. $\quad\square$

**Lemma 10.8.** *For any real* $m \times n$ *matrix* $\mathbf{A}$*, and any* $k \leq \min\{m,\ n\}$,

$$\max_{\substack{\mathbf{Y} \in \mathbb{R}^{n \times k} \\ \mathbf{Y}^\top \mathbf{Y} = \mathbf{I}_k}} \|\mathbf{AY}\|_{\mathrm{F}} = \left(\sum_{i=1}^k \sigma_i^2(\mathbf{A})\right)^{1/2}.$$

*The maximum is attained by* $\mathbf{Y}$ *coinciding with the* $k$ *leading right singular vectors of* $\mathbf{A}$.

*Proof.* Let $\mathbf{U\Sigma V}^\top$ be the singular value decomposition of $\mathbf{A}$; $\mathbf{U}$ and $\mathbf{V}$ are $m \times m$ and $n \times n$ unitary matrices respectively, while $\mathbf{\Sigma}$ is a diagonal matrix with $\Sigma_{jj} = \sigma_j$, the $j$th largest singular value of $\mathbf{A}$, $j = 1, \ldots, d$, where $d \triangleq \min\{m, n\}$. Due to the invariance of the Frobenius norm under unitary multiplication,

$$\|\mathbf{AY}\|_{\mathrm{F}}^2 = \|\mathbf{U\Sigma V}^\top \mathbf{Y}\|_{\mathrm{F}}^2 = \|\mathbf{\Sigma V}^\top \mathbf{Y}\|_{\mathrm{F}}^2. \tag{35}$$

Continuing from (35),

$$\|\mathbf{\Sigma V}^\top \mathbf{Y}\|_{\mathrm{F}}^2 = \mathrm{TR}\left(\mathbf{Y}^\top \mathbf{V\Sigma}^2 \mathbf{V}^\top \mathbf{Y}\right) = \sum_{i=1}^k \mathbf{y}_i^\top \left(\sum_{j=1}^d \sigma_j^2 \cdot \mathbf{v}_j \mathbf{v}_j^\top\right) \mathbf{y}_i = \sum_{j=1}^d \sigma_j^2 \cdot \sum_{i=1}^k \left(\mathbf{v}_j^\top \mathbf{y}_i\right)^2.$$

Let $z_j \triangleq \sum_{i=1}^k \left(\mathbf{v}_j^\top \mathbf{y}_i\right)^2$, $j = 1, \ldots, d$. Note that each individual $z_j$ satisfies

$$0 \leq z_j \triangleq \sum_{i=1}^k \left(\mathbf{v}_j^\top \mathbf{y}_i\right)^2 \leq \|\mathbf{v}_j\|^2 = 1,$$

where the last inequality follows from the fact that the columns of $\mathbf{Y}$ are orthonormal. Further,

$$\sum_{j=1}^d z_j = \sum_{j=1}^d \sum_{i=1}^k \left(\mathbf{v}_j^\top \mathbf{y}_i\right)^2 = \sum_{i=1}^k \sum_{j=1}^d \left(\mathbf{v}_j^\top \mathbf{y}_i\right)^2 = \sum_{i=1}^k \|\mathbf{y}_i\|^2 = k.$$

Combining the above, we conclude that

$$\|\mathbf{AY}\|_{\mathrm{F}}^2 = \sum_{j=1}^d \sigma_j^2 \cdot z_j \leq \sigma_1^2 + \ldots + \sigma_k^2. \tag{36}$$

Finally, it is straightforward to verify that if $\mathbf{y}_i = \mathbf{v}_i$, $i = 1, \ldots, k$, then (36) holds with equality. $\quad\square$

**Lemma 10.9.** *For any real* $d \times n$ *matrix* $\mathbf{A}$*, and pair of* $d \times k$ *matrix* $\mathbf{X}$ *and* $n \times k$ *matrix* $\mathbf{Y}$ *such that* $\mathbf{X}^\top \mathbf{X} = \mathbf{I}_k$ *and* $\mathbf{Y}^\top \mathbf{Y} = \mathbf{I}_k$ *with* $k \leq \min\{d,\ n\}$*, the following holds:*

$$\left|\mathrm{TR}\left(\mathbf{X}^\top \mathbf{AY}\right)\right| \leq \sqrt{k} \cdot \left(\sum_{i=1}^k \sigma_i^2(\mathbf{A})\right)^{1/2}.$$

*Proof.* By Lemma 10.7,

$$|\langle \mathbf{X},\ \mathbf{AY} \rangle| = \left|\mathrm{TR}\left(\mathbf{X}^\top \mathbf{AY}\right)\right| \leq \|\mathbf{X}\|_{\mathrm{F}} \cdot \|\mathbf{AY}\|_{\mathrm{F}} = \sqrt{k} \cdot \|\mathbf{AY}\|_{\mathrm{F}}.$$

where the last inequality follows from the fact that $\|\mathbf{X}\|_{\mathrm{F}}^2 = \mathrm{TR}\left(\mathbf{X}^\top \mathbf{X}\right) = \mathrm{TR}(\mathbf{I}_k) = k$. Combining with a bound on $\|\mathbf{AY}\|_{\mathrm{F}}$ as in Lemma 10.8, completes the proof. $\quad\square$

**Lemma 10.10.** *For any real* $d \times d$ *PSD matrix* $\mathbf{A}$*, and* $k \times d$ *matrix* $\mathbf{X}$ *with* $k \leq d$ *orthonormal columns,*

$$\mathrm{TR}\left(\mathbf{X}^\top \mathbf{AX}\right) \leq \sum_{i=1}^k \lambda_i(\mathbf{A})$$

*where* $\lambda_i(\mathbf{A})$ *is the* $i$th *largest eigenvalue of* $\mathbf{A}$*. Equality is achieved for* $\mathbf{X}$ *coinciding with the* $k$ *leading eigenvectors of* $\mathbf{A}$.

*Proof.* Let $\mathbf{A} = \mathbf{VV}^\top$ be a factorization of the PSD matrix $\mathbf{A}$. Then, $\mathrm{TR}\left(\mathbf{X}^\top \mathbf{AX}\right) = \mathrm{TR}\left(\mathbf{X}^\top \mathbf{VV}^\top \mathbf{X}\right) = \|\mathbf{V}^\top \mathbf{X}\|_{\mathrm{F}}^2$. The desired result follows by Lemma 10.8 and the fact that $\lambda_i(\mathbf{A}) = \sigma_i^2(\mathbf{V})$, $i = 1, \ldots, d$. $\quad\square$

# 11  Additional Experimental Results

Figure 3: Cumulative variance captured by $k$ $s$-spars components computed on the word-by-word matrix – BAGOFWORDS:NIPS dataset [30]. Sparsity is arbitrarily set to $s = 10$ nonzero entries per component. Fig. 3(a) depicts the cum. variance captured by $k = 6$ components. Deflation leads to a greedy formation of components; first components capture high variance, but subsequent ones contribute less. On the contrary, our algorithm jointly optimizes the $k$ components and achieves higher total cum. variance. Fig. 3(b) depicts the total cum. variance achieved for various values of $k$. Our algorithm operates on a rank-4 approximation of the input.

| | | Topic 1 | Topic 2 | Topic 3 | Topic 4 | Topic 5 | Topic 6 | Topic 7 | Topic 8 |
|---|---|---|---|---|---|---|---|---|---|
| | 1: | network | algorithm | neuron | parameter | object | classifier | word | noise |
| | 2: | model | data | cell | point | image | net | speech | control |
| | 3: | learning | system | pattern | distribution | recognition | classification | level | dynamic |
| TPOWER | 4: | input | error | layer | hidden | images | class | context | step |
| | 5: | function | weight | information | space | task | test | hmm | term |
| | 6: | neural | problem | signal | gaussian | features | order | character | optimal |
| | 7: | unit | result | visual | linear | feature | examples | processing | component |
| | 8: | set | number | field | probability | representation | rate | non | equation |
| | 9: | training | method | synaptic | mean | performance | values | approach | single |
| | 10: | output | vector | firing | case | view | experiment | trained | analysis |
| | 11: | network | algorithm | neuron | parameter | recognition | control | classifier | noise |
| | 12: | model | data | cell | distribution | object | action | classification | order |
| | 13: | input | weight | pattern | point | image | dynamic | class | term |
| SPANSPCA | 14: | learning | error | layer | linear | word | step | net | component |
| | 15: | neural | problem | signal | probability | performance | optimal | test | rate |
| | 16: | function | output | information | space | task | policy | speech | equation |
| | 17: | unit | result | visual | gaussian | features | states | examples | single |
| | 18: | set | number | synaptic | hidden | representation | reinforcement | approach | analysis |
| | 19: | system | method | field | case | feature | values | experiment | large |
| | 20: | training | vector | response | mean | images | controller | trained | form |
| | 21: | data | function | neuron | unit | learning | network | model | training |
| | 22: | distribution | algorithm | cell | weight | space | input | parameter | hidden |
| | 23: | gaussian | set | visual | layer | action | neural | information | performance |
| SPCABIPART | 24: | probability | error | direction | net | order | system | control | recognition |
| | 25: | component | problem | firing | task | step | output | dynamic | classifier |
| | 26: | approach | result | synaptic | connection | linear | pattern | mean | test |
| | 27: | analysis | number | response | activation | case | signal | noise | word |
| | 28: | mixture | method | spike | architecture | values | processing | field | speech |
| | 29: | likelihood | vector | activity | generalization | term | image | local | classification |
| | 30: | experiment | point | motion | threshold | optimal | object | equation | trained |

| | Total Cum. Variance |
|---|---|
| TPOWER | $2.5999 \cdot 10^3$ |
| SPANSPCA | $2.5981 \cdot 10^3$ |
| SPCABIPART | $\mathbf{3.2090 \cdot 10^3}$ |

Table 4: BAGOFWORDS:NIPS dataset [30]. We run various SPCA algorithms for $k = 8$ components (topics) and $s = 10$ nonzero entries per component. The table lists the words selected by each component (words corresponding to higher magnitude entries appear higher in the topic). Our algorithm was configured to use a rank-4 approximation of the input data.

| | Topic 1 | Topic 2 | Topic 3 | Topic 4 | Topic 5 | Topic 6 | Topic 7 | Topic 8 |
|---|---|---|---|---|---|---|---|---|
| **TPower** | | | | | | | | |
| 1: | percent | zzz_bush | team | school | women | zzz_enron | drug | palestinian |
| 2: | company | zzz_al_gore | game | student | show | firm | patient | zzz_israel |
| 3: | million | president | season | program | book | zzz_arthur_andersen | doctor | zzz_israeli |
| 4: | companies | official | player | high | com | deal | system | zzz_yasser_arafat |
| 5: | market | zzz_george_bush | play | children | look | lay | problem | attack |
| 6: | stock | campaign | games | right | american | financial | law | leader |
| 7: | business | government | point | group | need | energy | care | peace |
| 8: | money | plan | run | home | part | executives | cost | israelis |
| 9: | billion | administration | coach | public | family | accounting | help | israeli |
| 10: | fund | zzz_white_house | win | teacher | found | partnership | health | zzz_west_bank |
| **SpanSPCA** | | | | | | | | |
| 11: | percent | team | zzz_bush | palestinian | school | cup | show | won |
| 12: | company | game | zzz_al_gore | attack | student | minutes | com | night |
| 13: | million | season | president | zzz_united_states | children | add | part | left |
| 14: | companies | player | zzz_george_bush | zzz_u_s | program | tablespoon | look | big |
| 15: | market | play | campaign | military | home | teaspoon | need | put |
| 16: | stock | games | official | leader | family | oil | book | win |
| 17: | business | point | government | zzz_israel | women | pepper | called | hit |
| 18: | money | run | political | zzz_american | public | water | hour | job |
| 19: | billion | right | election | war | high | large | american | ago |
| 20: | plan | coach | group | country | law | sugar | help | zzz_new_york |
| **SPCABiPart** | | | | | | | | |
| 21: | percent | zzz_united_states | zzz_bush | company | team | cup | school | zzz_al_gore |
| 22: | million | zzz_u_s | official | companies | game | minutes | student | zzz_george_bush |
| 23: | money | zzz_american | government | market | season | add | children | campaign |
| 24: | high | attack | president | stock | player | tablespoon | women | election |
| 25: | program | military | group | business | play | oil | show | plan |
| 26: | number | palestinian | leader | billion | point | teaspoon | book | tax |
| 27: | need | war | country | analyst | run | water | family | public |
| 28: | part | administration | political | firm | right | pepper | look | zzz_washington |
| 29: | problem | zzz_white_house | american | sales | home | large | hour | member |
| 30: | com | games | law | cost | won | food | small | nation |

| | Total Cum. Variance |
|---|---|
| TPOWER | 45.4014 |
| SPANSPCA | 46.0075 |
| SPCABiPART | **47.7212** |

Table 5: BAGOFWORDS:NYTIMES dataset [30]. We run various SPCA algorithms for $k = 8$ components (topics) and $s = 10$ nonzero entries per component. The table lists the words selected by each component (words corresponding to higher magnitude entries appear higher in the topic). Our algorithm was configured to use a rank-$4$ approximation of the input data.

| | Topic 1 | Topic 2 | Topic 3 | Topic 4 | Topic 5 | Topic 6 | Topic 7 | Topic 8 |
|---|---|---|---|---|---|---|---|---|
| **TPOWER** | | | | | | | | |
| 1: | percent | zzz_bush | team | school | com | zzz_enron | law | palestinian |
| 2: | company | zzz_al_gore | game | student | women | firm | drug | zzz_israel |
| 3: | million | zzz_george_bush | season | program | book | deal | court | zzz_israeli |
| 4: | companies | campaign | player | children | web | financial | case | zzz_yasser_arafat |
| 5: | market | right | play | show | american | zzz_arthur_andersen | federal | peace |
| 6: | stock | group | games | public | information | chief | patient | israelis |
| 7: | money | political | point | need | look | executive | system | israeli |
| 8: | business | zzz_united_states | run | part | site | analyst | decision | military |
| 9: | government | zzz_u_s | coach | family | zzz_new_york | executives | bill | zzz_palestinian |
| 10: | official | administration | home | help | question | lay | member | zzz_west_bank |
| 11: | billion | leader | win | job | number | investor | lawyer | war |
| 12: | president | attack | won | teacher | called | energy | doctor | security |
| 13: | plan | zzz_white_house | night | country | find | investment | cost | violence |
| 14: | high | tax | left | problem | found | employees | care | killed |
| 15: | fund | zzz_washington | guy | parent | ago | accounting | health | talk |
| **SPANSPCA** | | | | | | | | |
| 16: | percent | team | official | zzz_al_gore | cup | show | public | night |
| 17: | company | game | zzz_bush | zzz_george_bush | minutes | com | member | big |
| 18: | million | season | zzz_united_states | campaign | add | part | system | set |
| 19: | companies | player | attack | election | tablespoon | look | case | film |
| 20: | market | play | zzz_u_s | political | teaspoon | need | number | find |
| 21: | stock | games | palestinian | vote | oil | book | question | room |
| 22: | business | point | military | republican | pepper | women | job | place |
| 23: | money | run | leader | voter | water | family | told | friend |
| 24: | billion | right | zzz_american | democratic | large | called | put | took |
| 25: | plan | win | war | school | sugar | children | zzz_washington | start |
| 26: | government | coach | zzz_israel | presidential | serving | help | found | car |
| 27: | president | home | country | zzz_white_house | butter | ago | information | feel |
| 28: | high | won | administration | law | chopped | zzz_new_york | federal | half |
| 29: | cost | left | terrorist | zzz_republican | hour | program | student | guy |
| 30: | group | hit | american | tax | pan | problem | court | early |
| **SPCABIPART** | | | | | | | | |
| 31: | company | show | cup | team | percent | zzz_al_gore | official | school |
| 32: | companies | home | minutes | game | million | zzz_george_bush | zzz_bush | student |
| 33: | stock | run | add | season | money | campaign | government | children |
| 34: | market | com | tablespoon | player | plan | right | president | women |
| 35: | billion | high | oil | play | business | election | zzz_united_states | book |
| 36: | zzz_enron | need | teaspoon | games | tax | political | zzz_u_s | family |
| 37: | firm | look | pepper | coach | cost | point | group | called |
| 38: | analyst | part | water | guy | cut | leader | attack | hour |
| 39: | industry | night | large | yard | job | zzz_washington | zzz_american | friend |
| 40: | fund | zzz_new_york | sugar | hit | pay | administration | country | found |
| 41: | investor | help | serving | played | deal | question | military | find |
| 42: | sales | left | butter | playing | quarter | member | american | set |
| 43: | customer | put | chopped | ball | chief | won | war | room |
| 44: | investment | ago | fat | fan | executive | win | law | film |
| 45: | economy | big | food | shot | financial | told | public | small |

| | Total Cum. Variance |
|---|---|
| TPOWER | 48.140645 |
| SPANSPCA | 48.767864 |
| SPCABIPART | **51.873063** |

Table 6: BAGOFWORDS:NYTIMES dataset [30]. We run various SPCA algorithms for $k = 8$ components (topics) and cardinality $s = 15$ per component. The table lists the words corresponding to each component (words corresponding to higher magnitude entries appear higher in the topic). Our algorithm was configured to use a rank-4 approximation of the input data.

| | Topic 1 | Topic 2 | Topic 3 | Topic 4 | Topic 5 | Topic 6 | Topic 7 | Topic 8 |
|---|---|---|---|---|---|---|---|---|
| 1: | percent | zzz_bush | team | school | com | zzz_enron | drug | palestinian |
| 2: | company | zzz_al_gore | game | student | women | court | patient | zzz_israel |
| 3: | million | zzz_george_bush | season | program | book | case | doctor | zzz_israeli |
| 4: | companies | campaign | player | children | web | firm | cell | zzz_yasser_arafat |
| 5: | market | zzz_united_states | play | show | site | federal | care | peace |
| 6: | stock | zzz_u_s | games | public | information | lawyer | disease | israelis |
| 7: | government | political | point | part | zzz_new_york | deal | health | israeli |
| 8: | official | attack | run | family | www | decision | medical | zzz_palestinian |
| 9: | money | zzz_american | home | system | hour | chief | test | zzz_west_bank |
| 10: | business | american | coach | help | find | power | hospital | security |
| 11: | president | administration | win | problem | mail | industry | research | violence |
| 12: | billion | leader | won | law | found | executive | cancer | killed |
| 13: | plan | country | left | job | put | according | treatment | talk |
| 14: | group | election | night | called | set | financial | study | meeting |
| 15: | high | zzz_washington | hit | look | room | office | death | soldier |
| 16: | right | military | guy | member | big | analyst | human | minister |
| 17: | fund | zzz_white_house | yard | question | told | executives | heart | zzz_sharon |
| 18: | need | war | played | ago | friend | zzz_arthur_andersen | blood | fire |
| 19: | cost | tax | start | teacher | director | employees | trial | zzz_ariel_sharon |
| 20: | number | nation | playing | parent | place | investor | benefit | zzz_arab |
| 21: | percent | team | zzz_al_gore | attack | school | cup | com | drug |
| 22: | company | game | zzz_bush | zzz_united_states | student | minutes | web | patient |
| 23: | million | season | zzz_george_bush | zzz_u_s | children | add | site | cell |
| 24: | companies | player | campaign | palestinian | program | tablespoon | information | doctor |
| 25: | market | play | election | military | family | oil | computer | disease |
| 26: | stock | games | political | zzz_american | women | teaspoon | find | care |
| 27: | business | point | tax | zzz_israel | show | pepper | big | health |
| 28: | money | run | republican | war | help | water | zzz_new_york | test |
| 29: | billion | win | zzz_white_house | country | told | large | www | research |
| 30: | government | home | vote | terrorist | parent | sugar | mail | human |
| 31: | president | won | law | american | problem | serving | set | medical |
| 32: | plan | coach | administration | zzz_taliban | book | butter | put | study |
| 33: | high | left | democratic | zzz_afghanistan | job | chopped | director | death |
| 34: | group | night | voter | security | found | hour | industry | cancer |
| 35: | official | hit | leader | zzz_israeli | friend | pan | room | hospital |
| 36: | need | guy | public | nation | ago | fat | small | treatment |
| 37: | right | yard | zzz_republican | member | question | bowl | car | scientist |
| 38: | part | played | presidential | support | teacher | gram | zzz_internet | according |
| 39: | cost | look | federal | called | case | food | place | blood |
| 40: | system | start | zzz_washington | forces | number | medium | film | heart |
| 41: | palestinian | percent | zzz_al_gore | cup | school | team | company | official |
| 42: | zzz_israel | million | zzz_bush | minutes | right | game | companies | government |
| 43: | zzz_israeli | money | zzz_george_bush | add | group | season | market | president |
| 44: | zzz_yasser_arafat | billion | campaign | tablespoon | show | player | stock | zzz_united_states |
| 45: | peace | business | election | oil | home | play | zzz_enron | zzz_u_s |
| 46: | war | fund | political | teaspoon | high | games | analyst | attack |
| 47: | terrorist | tax | zzz_white_house | pepper | program | point | firm | zzz_american |
| 48: | zzz_taliban | cost | administration | water | need | run | industry | country |
| 49: | zzz_afghanistan | cut | republican | hour | part | coach | investor | law |
| 50: | forces | job | leader | large | com | win | sales | plan |
| 51: | bin | pay | vote | sugar | american | won | customer | public |
| 52: | troop | economy | democratic | serving | look | left | price | zzz_washington |
| 53: | laden | deal | presidential | butter | help | night | investment | member |
| 54: | student | big | zzz_clinton | chopped | problem | hit | quarter | system |
| 55: | zzz_pakistan | chief | support | pan | called | guy | executives | nation |
| 56: | product | executive | zzz_congress | fat | zzz_new_york | yard | consumer | case |
| 57: | zzz_internet | financial | military | bowl | number | played | technology | federal |
| 58: | profit | start | policy | gram | question | ball | share | information |
| 59: | earning | record | court | food | ago | playing | prices | power |
| 60: | shares | manager | security | league | told | lead | growth | effort |

(Rows 1–20: TPOWER; rows 21–40: SPANSPCA; rows 41–60: SPCABIPART)

| | Total Cum. Variance |
|---|---|
| TPOWER | 50.7686 |
| SPANSPCA | 52.8117 |
| SPCABIPART | **54.8906** |

Table 7: BAGOFWORDS:NYTIMES dataset [30]. We run various SPCA algorithms for $k = 8$ components (topics) and cardinality $s = 20$ per component. The table lists the words corresponding to each component (words corresponding to higher magnitude entries appear higher in the topic). Our algorithm was configured to use a rank-4 approximation of the input data.

Figure 4: Cumulative variance captured by $k$ $s$-sparse ($s = 10$) extracted components on the word-by-word matrix – BAGOFWORDS:NYTIMES dataset [30]. Fig. 4(a) depicts the cum. variance captured by $k = 6$ components. Deflation leads to a greedy formation of components; first components capture high variance, but subsequent ones contribute less. On the contrary, our algorithm jointly optimizes the $k$ components and achieves higher total cum. variance. Fig. 4(b) depicts the total cum. variance achieved for various values of $k$. Sparsity is arbitrarily set to $s = 10$ nonzero entries per component. Our algorithm operates on a rank-4 approximation.

Figure 5: Same as Fig. 4, but for sparsity $s = 15$.

Figure 6: Same as Fig. 4, but for sparsity $s = 20$.