[Reviews · NeurIPS 2015]

Submitted by Assigned_Reviewer_1

This paper presents a very elegant way of doing sparse PCA with disjoint supports. The algorithm is based on an epsilon-net and bipartite matching (for max-weigthed matching). The experiments show how much reliable is this approach.

Reading the paper is a pleasure and the algorithm is well described. Still I have some very minor remarks, 1) There is no section 7 in the paper, but it is in the supplementary material (please add a footnote). 2) Please in page 2 when you introduce PSD (I assume it is for Positive Semi-Definite) explains it as you did for PCA. 3) Page 5, do you really need to have almost a full sentence in italic ? (end of the first paragraph) 4) Perhaps you should cite "Structured Sparse Principal Component Analysis" by Jenatton et al, AISTATS 2010 as they propose structured sparsity (disjoint support is a kind of structure).

I have a quick question: Is there a way to approximate the computation of the epsilon-net and still having some guarantee ? It is to be the bottleneck of the approach.

Summary: This paper presents a very elegant way of doing sparse PCA with disjoint supports. Using a difference point of view than classical sparse analysis allows the authors to propose an effective sparse PCA algorithm than seems better than state-of-art approaches.

Submitted by Assigned_Reviewer_2

The paper proposed a new method for sparse PCA. The key is to generate sparse components via bipartite matchings. The problem studied in the paper is interesting and the idea proposed in the paper is intuitive. The paper is well written and easy to understand. My major concerns on the paper is listed below: 1. The authors propose to generate multiple disjoint components. Although this ensures that the generated components are orthogonal, it also makes the problem much complicated. The authors need to justify whether this is worthwhile. 2. To search for good X, the proposed algorithm needs to search on a net. When \epsilon is small, there can be too many points on the net and make the method impractical for large scale problem. 3. Authors needs to provide running time results of different algorithms on the benchmark data sets used in the experiment.

Summary: The paper proposed a new method for sparse PCA. The key is to generate sparse components via bipartite matchings.

Submitted by Assigned_Reviewer_3

I have read the paper "Sparse PCA via Bipartite matching".

The paper studies the problem of sparse PCA in the setting of multi-components with disjoint supports.

The paper is generally well-written.

The main idea is well presented and is generally easy to digest.

However, the major problem of the paper

is the focus on the case of

"disjoint components".

In detail,

in equation (2) and the line below, the paper requires all supports of X^{(i)} are disjoint with each other; all the methods theory are built upon such a constraint.

This raises serious doubt on how the algorithm can be used (or to be advantageous over existing ones),

especially

from a practical perspective.

Take gene microarray data

for example (the paper presents some examples on the microarray data),

it is believed that the support of different components are monotone: the support of genes differentially expressed between a normal patient and a patient with a malignant tumor is usually a subset of that between a normal patient and a patient with a malicious tumor. In other words, if we have two principal components $\mu_1$ and $\mu_2$, then it is more likely that

\[

\mathrm{supp}(\mu_1) \subset \mathrm{supp}(\mu_2),

\]

instead of

\[ \mathrm{supp}(\mu_1) \cap \mathrm{supp}(\mu_2)= \emptyset.

\] Even from a theoretical viewpoint,

assuming the supports are disjoint is rather strong.

While I appreciate the paper has made a fair contribution in terms of algorithm (which is

not that significant though,

given the existing literature on sparse PCA), I think the paper lacks a good motivation, which largely down-weights the importance of the paper.

To patch up such a weakness, I would suggest the authors make a serious discussion on the following problems, especially in methods and theory.

\begin{itemize}

\item How broad is the model in equation (2) and the line below.

\item In case of overlapping supports, how would we

deal with it? If theres is

a

convenient extension of the presented

approach, would the promised advantages continue to exist?

\end{itemize}

Overall, while I like some of ideas in

the paper, I think there are serious drawbacks. Based on these, I would like to recommend rejection with resubmission.

Summary: The authors need to make the motivation of the study more convincing.

Author Feedback
Author rebuttal: We thank the reviewers for their constructive comments and positive assessment of our work.
The major criticism came from Reviewers 3 and 4 and considered the usefulness of the disjoint supports assumption.

We would like to emphasize that disjoint supports is a desired structure; not just a means towards achieving orthogonality.
Specifically, in the case of a text data, extracted PCs with disjoint supports can be interpreted as different topics defined by disjoint sets of words. In this case, simple orthogonality has no operational meaning.

However, the usefulness of a particular structure in the constraints heavily depends on the problem at hand. As Reviewer 4 correctly notes, disjointness is not a suitable structure for the microarray example.

We would like to stress out that in the Sparse PCA literature there is very limited work on *multiple* components (See "Related Work" section).
Some approaches do not enforce any constraint "allowing" for overlapping supports, while others assume that the desired components share exactly the same nonzero entries.
Most importantly, there are no previous theoretical results for multiple components with disjoint supports; in the literature it is typically suggested that disjoint components can be greedily computed using deflation. In our paper, we show that this is suboptimal and design the first algorithm that *jointly* selects multiple sparse eigenvectors and has provable performance bounds.

Detailed responses follow.
We note that we have corrected typos, misconceptions and notation issues.

REVIEWER #1:

1. [...] I have some very minor remarks, [...]

We will address the proposed changes in the final manuscript.

2. Is there a way to approximate the computation of the epsilon-net and still having some guarantee ? [...]

We agree that the cardinality of the epsilon net is a computational bottleneck; for our theoretical guarantees its size grows exponentially in the rank of the input. Improving this dependence or providing necessary lower bounds is indeed a very interesting question.

REVIEWER #3:

1. To search for good X, the proposed algorithm needs to search on a net. [...] there can be too many points on the net [...]

The cardinality of the epsilon net is indeed a computational bottleneck. However, that simple construction allows us to obtain the first non-trivial algorithm for the Sparse PCA problem with multiple disjoint components, which is also accompanied by theoretical approximation guarantees. Improving this complexity is an interesting research direction. In practice, the points of the net are (independently) generated one by one, and our algorithm achieves competitive results considering only a relatively small number of points.

2. Authors needs to provide running time results of different algorithms [...]

We will address this comment in the final manuscript. In our experimental evaluation our algorithm is terminated early. It is slower compared to the iterative deflation-based approaches, but all methods are executed for a few minutes.

REVIEWER #4:

1.[...] To patch up such a weakness, I would suggest the authors make a serious discussion on the following problems [...]
- How broad is the model in equation (2) [...]?
- In case of overlapping supports, how would we deal with it? If there is a convenient extension of the presented approach, would the promised advantages continue to exist?

We will take these constructive suggestions into account. With this opportunity we note the following: the bipartite-matching solver which solves Eq (6) roughly "projects" a matrix on the combinatorial constraint set of disjoint supports. Our approach can be extended to other structural constraints as long as we can design a solver for Eq. (6) for those constraints, and similar theoretical guarantees can be obtained.